# Conditional Image Generation by Conditioning Variational Auto-Encoders

**William Harvey, Saeid Naderiparizi & Frank Wood**[*]
Department of Computer Science
University of British Columbia
Vancouver, Canada
`{wsgh,saeidnp,fwood}@cs.ubc.ca`

## Abstract

We present a conditional variational auto-encoder (VAE) which, to avoid the substantial cost of training from scratch, uses an architecture and training objective capable of leveraging a foundation model in the form of a pretrained unconditional VAE. To train the conditional VAE, we only need to train an artifact to perform amortized inference over the unconditional VAE's latent variables given a conditioning input. We demonstrate our approach on tasks including image inpainting, for which it outperforms state-of-the-art GAN-based approaches at faithfully representing the inherent uncertainty. We conclude by describing a possible application of our inpainting model, in which it is used to perform Bayesian experimental design for the purpose of guiding a sensor.

## 1 Introduction

A major challenge with applying variational auto-encoders (VAEs) to high-dimensional data is the typically slow training times. For example, training a state-of-the-art VAE (Vahdat & Kautz, 2020; Child, 2020) on the $256 \times 256$ FFHQ dataset (Karras et al., 2019) takes on the order of 1 GPU-year, but a state-of-the-art generative adversarial network (GAN) (Lin et al., 2021; Karras et al., 2020) can be trained on the same dataset in a matter of GPU-weeks. One hypothesis for the cause of this disparity is that, whereas the "mass-covering" training objective for a VAE forces it to assign probability mass over the entirety of the data distribution, a GAN can "cut corners" by dropping modes (Arora & Zhang, 2017; Arora et al., 2017).

We focus on the problem of *conditional* generative modelling: given an input (e.g. a partially blanked-out image), we wish to map to a distribution over outputs (e.g. plausible completions of the image). Both conditional GANs (Zheng et al., 2019; Zhao et al., 2021) and conditional VAEs (Sohn

---

[*]Frank Wood is also affiliated with the Montréal Institute for Learning Algorithms (Mila) and Inverted AI.

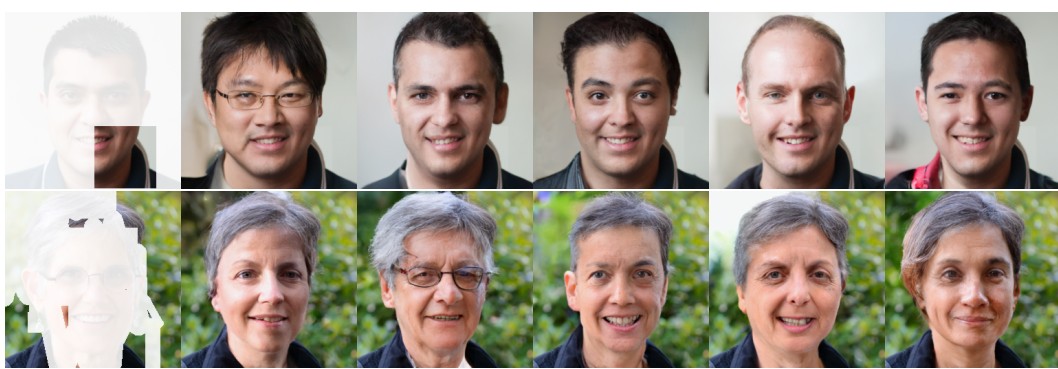

Figure 1: **Left column:** Images with most pixels masked out. **Rest:** Completions from our method.

et al., 2015; Ivanov et al., 2018) are applicable to this problem, with the same disparity in training times that we described for their unconditional counterparts. We present an approach based on the conditional VAE framework but, to mitigate the associated slow training times, we design the architecture so that we can incorporate pretrained unconditional VAEs. We show that re-using publicly available pretrained models in this way can lead to training times and sample quality competitive with GANs, while avoiding mode dropping.

While requiring an existing pretrained model is a limitation, we note that: **(I)** The unconditional VAE need not have been (pre-)trained on the same dataset as the conditional model; we show unconditional models trained on ImageNet are suitable for later use with various photo datasets. **(II)** A single unconditional VAE can be used for later training of conditional VAEs on any desired conditional generation tasks (e.g. the same image model may be later used for image completion or image colourisation). **(III)** There is an increasing trend in the machine learning community towards sharing large, expensively trained models (Wolf et al., 2020), sometimes referred to as foundation models (Bommasani et al., 2021). Most of the unconditional VAEs in our experiments use publicly-available pretrained weights released by Child (2020). By presenting a use case for foundation models in image modelling, we hope to encourage even more sharing of pretrained weights in this domain.

We demonstrate our approach on several conditional generation tasks in the image domain but focus in particular on stochastic image completion: the problem of inferring the posterior distribution over images given the observation of a subset of pixel values. For some applications such as photo-editing the implicit distribution defined by GANs is good enough. We argue that our approach has substantial advantages when image completion is used as part of a larger pipeline, and discuss one possible instance of this in Section 5: Bayesian optimal experimental design (BOED) for guiding a sensor or hard attention mechanism (Ma et al., 2018; Harvey et al., 2019; Rangrej & Clark, 2021). In this case, missing modes of the posterior over images is likely to lead to bad decisions. We show that our objective corresponds to the mass-covering KL divergence and so covers the posterior well. This is supported empirically by results indicating that, not only is the visual quality of our image completions (see Fig. 1) close to the state-of-the-art (Zhao et al., 2021), but our coverage of the "true" posterior over image completions is superior to that of any of our baselines.

**Contributions** We develop a method to cheaply convert pretrained unconditional VAEs into conditional VAEs. The resulting training times and sample quality are competitive with GANs, while the models avoid the mode-dropping behaviour associated with GANs. Finally, we showcase a possible application in Bayesian optimal experimental design that benefits from these capabilities.

## 2 VARIATIONAL AUTO-ENCODERS

We describe VAEs in terms of three components. **(I)** A decoder with parameters $\theta \in \Theta$ maps from latent variables $\mathbf{z}$ to a distribution over data $\mathbf{x}$, which we call $p_{\text{model}}(\mathbf{x}|\mathbf{z};\theta)$. **(II)** There is a prior over latent variables, $p_{\text{model}}(\mathbf{z};\theta)$. This may have learnable parameters, which we consider to be part of $\theta$. Together, the prior and decoder define a joint distribution, $p_{\text{model}}(\mathbf{z},\mathbf{x};\theta)$. Finally, **(III)** an encoder with parameters $\phi \in \Phi$ maps from data to an approximate posterior distribution over latent variables, $q(\mathbf{z}|\mathbf{x};\phi) \approx p_{\text{model}}(\mathbf{z}|\mathbf{x};\theta)$. Ideally, $\theta$ would be learned to maximise the log likelihood $\log p_{\text{model}}(\mathbf{x};\theta) = \log \int p_{\text{model}}(\mathbf{z},\mathbf{x};\theta)d\mathbf{z}$, averaged over training examples. Since this is intractable, $\theta$ and $\phi$ are instead trained jointly to maximise an average of the evidence lower-bound (ELBO) over each training example $\mathbf{x} \sim p_{\text{data}}(\cdot)$:

$$\mathbb{E}_{p_{\text{data}}(\mathbf{x})}\left[\text{ELBO}(\theta,\phi,\mathbf{x})\right] = \mathbb{E}_{p_{\text{data}}(\mathbf{x})}\mathbb{E}_{q(\mathbf{z}|\mathbf{x};\phi)}\left[\log \frac{p_{\text{model}}(\mathbf{z};\theta)p_{\text{model}}(\mathbf{x}|\mathbf{z};\theta)}{q(\mathbf{z}|\mathbf{x};\phi)}\right] \quad (1)$$

$$= -\mathcal{H}\left[p_{\text{data}}(\mathbf{x})\right] - \text{KL}\big(p_{\text{data}}(\mathbf{x})q(\mathbf{z}|\mathbf{x};\phi) \,\|\, p_{\text{model}}(\mathbf{z},\mathbf{x};\theta)\big). \quad (2)$$

The data distribution's entropy, $\mathcal{H}\left[p_{\text{data}}(\mathbf{x})\right]$, is typically a finite constant, and this is guaranteed in our experiments where $\mathbf{x}$ is an image with discrete pixel values. Maximising the above objective will therefore drive $p_{\text{model}}(\mathbf{z},\mathbf{x};\theta)$ towards $p_{\text{data}}(\mathbf{x})q(\mathbf{z}|\mathbf{x};\phi)$, and so the marginal $p_{\text{model}}(\mathbf{x};\theta)$ towards $p_{\text{data}}(\mathbf{x})$. The KL divergence shown leads to mass-covering behaviour from $p_{\text{model}}(\mathbf{z},\mathbf{x};\theta)$ (Bishop, 2006) so $p_{\text{model}}(\mathbf{x};\theta)$ should assign probability broadly over the data distribution $p_{\text{data}}(\mathbf{x})$. For notational simplicity in the rest of the paper, parameters $\theta$ and $\phi$ are not written when clear from the context.

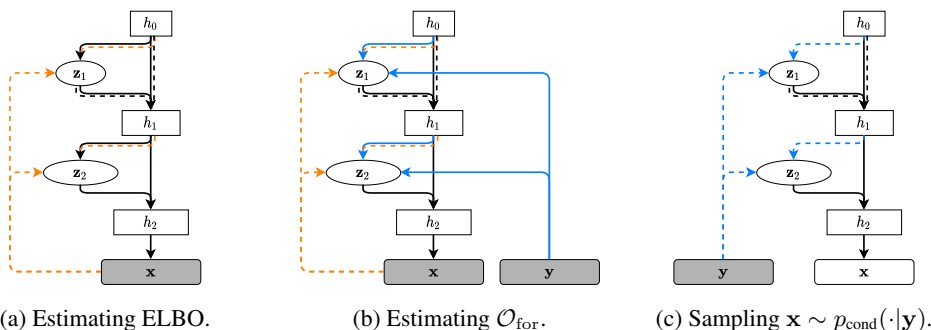

(a) Estimating ELBO.    (b) Estimating $\mathcal{O}_{\text{for}}$.    (c) Sampling $\mathbf{x} \sim p_{\text{cond}}(\cdot|\mathbf{y})$.

Figure 2: A hierarchical VAE architecture with $L = 2$ layers of latent variables. Part (a) illustrates the computation of the ELBO for an unconditional VAE; part (b) illustrates the computation of our training objective $\mathcal{O}_{\text{for}}$; and part (c) illustrates the drawing of conditional samples. The encoder is shown in orange; the prior and the decoder (which maintains a deterministic hidden state $h_l$) are both shown in black; and the partial encoder is shown in blue. The computation graph needed to sample $\mathbf{z}$ in each case is drawn with dashed lines, and the remainder of the computation graph is drawn with solid lines.

Hierarchical VAEs (Gregor et al., 2015; Kingma et al., 2016; Sønderby et al., 2016; Klushyn et al., 2019) partition the latent variables $\mathbf{z}$ in a way which has been found to improve the fidelity of the learned $p_{\text{model}}(\mathbf{x})$, especially for the image domain (Vahdat & Kautz, 2020; Child, 2020). In particular, they define $\mathbf{z}$ to consist of $L$ disjoint groups, $\mathbf{z}_1, \ldots, \mathbf{z}_L$. The prior for each $\mathbf{z}_l$ can depend on the previous groups through the factorisation

$$p_{\text{model}}(\mathbf{z}) = \prod_{l=1}^{L} p_{\text{model}}(\mathbf{z}_l|\mathbf{z}_{<l}). \tag{3}$$

where $\mathbf{z}_{<l}$ is the null set for $l = 1$ and $\{\mathbf{z}_1, \ldots, \mathbf{z}_{l-1}\}$ otherwise. Fig. 2a shows the hierarchical VAE architecture we base this work on, in which the dependency of the prior for each $\mathbf{z}_l$ on $\mathbf{z}_{<l}$ is maintained via the decoder's hidden state $h_l$. The distribution produced by the encoder for each $\mathbf{z}_l$ also depends on the previous hidden state $h_{l-1}$ and so factorises as $q(\mathbf{z}|\mathbf{x}) = \prod_{l=1}^{L} q(\mathbf{z}_l|\mathbf{z}_{<l}, \mathbf{x})$. We will parameterise $p_{\text{model}}(\mathbf{z}_l|\mathbf{z}_{<l})$ and $q(\mathbf{z}_l|\mathbf{z}_{<l}, \mathbf{x})$ as diagonal Gaussian distributions, as is common for hierarchical VAEs (Sønderby et al., 2016; Vahdat & Kautz, 2020; Child, 2020).

## 3    AMORTIZED INFERENCE IN A PRETRAINED ARTIFACT

To convert an unconditional VAE architecture to a conditional architecture, we introduce a *partial encoder* with parameters $\psi \in \Psi$. Given a conditioning input $\mathbf{y}$ (which could be, e.g., an image with some pixels masked out in the case of inpainting), the partial encoder defines an approximate posterior over the latent variables, $c(\mathbf{z}|\mathbf{y}; \psi)$. Since the conditional generation task is to approximate $p_{\text{data}}(\mathbf{x}|\mathbf{y})$, we use the partial encoder to define

$$p_{\text{cond}}(\mathbf{x}|\mathbf{y}; \theta, \psi) := \int p_{\text{model}}(\mathbf{x}|\mathbf{z}; \theta) c(\mathbf{z}|\mathbf{y}; \psi) \mathrm{d}\mathbf{z} \tag{4}$$

with learnable parameters $\theta$ and $\psi$. We can sample from $p_{\text{cond}}(\mathbf{x}|\mathbf{y})$ by sampling $\mathbf{z} \sim c(\cdot|\mathbf{y})$ and then $\mathbf{x} \sim p_{\text{model}}(\cdot|\mathbf{z})$ as shown in Fig. 2c. This defines a conditional VAE architecture which, unique amongst related work with high-dimensional $\mathbf{x}$ and $\mathbf{y}$ (Sohn et al., 2015; Zheng et al., 2019; Ivanov et al., 2018; Wan et al., 2021), has a decoder $p_{\text{model}}(\mathbf{x}|\mathbf{z}; \theta)$ with no dependence on $\mathbf{y}$. This decoder can therefore use an architecture identical to that of an unconditional VAE and also, as we will show later, re-use unconditional VAE weights.

We now introduce some notation before describing our method further. Let the distribution of paired data be $p_{\text{data}}(\mathbf{x}, \mathbf{y})$, and recall that training an unconditional VAE matches two joint distributions: the distribution of unconditional samples, $p_{\text{model}}(\mathbf{z}, \mathbf{x})$; and the distribution resulting from sampling data $\mathbf{x}$ and then encoding it, $p_{\text{data}}(\mathbf{x})q(\mathbf{z}|\mathbf{x})$. We define extensions of each to include $\mathbf{y}$:

$$p_{\text{model}}(\mathbf{z}, \mathbf{x}, \mathbf{y}; \theta) = p_{\text{model}}(\mathbf{z}; \theta)p_{\text{model}}(\mathbf{x}|\mathbf{z}; \theta)p_{\text{data}}(\mathbf{y}|\mathbf{x}), \tag{5}$$

$$r(\mathbf{z}, \mathbf{x}, \mathbf{y}; \phi) = p_{\text{data}}(\mathbf{x}, \mathbf{y})q(\mathbf{z}|\mathbf{x}; \phi), \tag{6}$$

where $p_{\text{data}}(\mathbf{y}|\mathbf{x})$ is a (potentially intractable) conditional distribution under $p_{\text{data}}(\mathbf{x}, \mathbf{y})$. Note that $p_{\text{model}}(\mathbf{z}, \mathbf{x}, \mathbf{y}; \theta)$ and $r(\mathbf{z}, \mathbf{x}, \mathbf{y}; \phi)$ are exactly the two distributions matched by the unconditional VAE objective in Eq. (2) with an additional factor of $p_{\text{data}}(\mathbf{y}|\mathbf{x})$. Therefore, if the unconditional VAE represented by $\theta$ and $\phi$ is well trained, $p_{\text{model}}(\mathbf{z}, \mathbf{x}, \mathbf{y}; \theta)$ and $r(\mathbf{z}, \mathbf{x}, \mathbf{y}; \phi)$ will be close. From now on, we will use $p_{\text{model}}$ and $r$ to refer to any marginals and conditionals of the above joint distributions, with the specific marginal or conditional clear from context.

## 3.1 TRAINING OBJECTIVE

Our training objective, previously used for training conditional VAEs (Sohn et al., 2015; Ivanov et al., 2018) and neural processes (Garnelo et al., 2018), is

$$\mathcal{O}_{\text{for}}(\theta, \phi, \psi) = \mathbb{E}_{p_{\text{data}}(\mathbf{x}, \mathbf{y})} \mathbb{E}_{q(\mathbf{z}|\mathbf{x})} \left[ \log \frac{p_{\text{model}}(\tilde{\mathbf{x}}|\mathbf{z}) c(\mathbf{z}|\mathbf{y})}{q(\mathbf{z}|\mathbf{x})} \right] \leq \mathbb{E}_{p_{\text{data}}(\mathbf{x}, \mathbf{y})} \left[ \log p_{\text{cond}}(\tilde{\mathbf{x}}|\mathbf{y}) \right] \quad (7)$$

where $\tilde{\mathbf{x}}$ is the part of $\mathbf{x}$ we wish to predict. In general we can set $\tilde{\mathbf{x}} := \mathbf{x}$ but for inpainting define $\tilde{\mathbf{x}}$ to consist of only the dimensions of $\mathbf{x}$ not observed in $\mathbf{y}$, abusing notation by ignoring the implication that $\tilde{\mathbf{x}}$ is formally a function of $\mathbf{y}$ as well as $\mathbf{x}$. Our architectures have pixel-wise independent likelihoods so $p_{\text{model}}(\tilde{\mathbf{x}}|\mathbf{z})$ is tractable in either case. Equation (7) lower-bounds $\log p_{\text{cond}}(\tilde{\mathbf{x}}|\mathbf{y})$ similarly to how the ELBO of an unconditional VAE lower-bounds $\log p_{\text{model}}(\mathbf{x})$. The major difference is that the prior, $p_{\text{model}}(\mathbf{z})$, is replaced by $c(\mathbf{z}|\mathbf{y})$. This is reflected in Fig. 2b where each $\mathbf{z}_l$ is conditioned on $\mathbf{y}$ via the partial encoder. Similar to standard estimators for an unconditional hierarchical VAE's ELBO, reduced-variance estimates of Eq. (7) can be obtained by computing KL divergences between $q(\mathbf{z}_l|\mathbf{z}_{<l}, \mathbf{x})$ and $c(\mathbf{z}_l|\mathbf{z}_{<l}, \mathbf{y})$ analytically for each layer $l$ (see Appendix F for details).

We are particularly interested in the properties of the learned partial encoder. Recall the joint distribution $r(\mathbf{z}, \mathbf{x}, \mathbf{y}; \phi) = p_{\text{data}}(\mathbf{x}, \mathbf{y}) q(\mathbf{z}|\mathbf{x}; \phi)$. Then $r(\mathbf{z}|\mathbf{y}; \phi)$ is the intractable posterior given by marginalising out $\mathbf{x}$ and conditioning on $\mathbf{y}$. We find that fitting $\psi$ to maximise $\mathcal{O}_{\text{for}}(\theta, \phi, \psi)$ is equivalent to minimising the mass-covering KL divergence from $r(\mathbf{z}|\mathbf{y}; \phi)$ to $c(\mathbf{z}|\mathbf{y}; \psi)$. We formalise this statement in the following theorem, which is proven in Appendix C.

**Theorem 3.1.** *For any set $\Psi$ of permissible values of $\psi$, and for any $\theta \in \Theta$ and $\phi \in \Phi$,*

$$\arg\max_{\psi \in \Psi} \mathcal{O}_{\text{for}}(\theta, \phi, \psi) = \arg\min_{\psi \in \Psi} \mathbb{E}_{p_{\text{data}}(\mathbf{y})} \left[ KL\big(r(\mathbf{z}|\mathbf{y}; \phi) \,\|\, c(\mathbf{z}|\mathbf{y}; \psi)\big) \right]. \quad (8)$$

Due to the mass-covering properties of this "forward" KL divergence (Bishop, 2006), this theorem indicates that the learned $c(\mathbf{z}|\mathbf{y}; \psi)$ should have good coverage of all modes of $r(\mathbf{z}|\mathbf{y}; \phi)$. Intuitively, the resulting diverse samples of latent variables $\mathbf{z} \sim c(\cdot|\mathbf{y}; \psi)$ should lead to diverse samples of $\tilde{\mathbf{x}} \sim p_{\text{model}}(\cdot|\mathbf{z})$ which cover the "true" posterior $p_{\text{data}}(\tilde{\mathbf{x}}|\mathbf{y})$. We formalise this in Appendix C by showing that maximising $\mathcal{O}_{\text{for}}$ also minimises an upper-bound on a KL divergence in $\tilde{\mathbf{x}}$-space.

## 3.2 FASTER TRAINING WITH A PRETRAINED VAE

To justify using weights trained as part of an unconditional VAE we present Theorem 3.2

**Theorem 3.2.** *Assume that we have a sufficiently expressive encoder and decoder that there exist parameters $\theta^* \in \Theta$ and $\phi^* \in \Phi$ which make the unconditional VAE objective (Eq. (1)) equal to its upper bound of $-\mathcal{H}\left[p_{\text{data}}(\mathbf{x})\right]$. Assume also that $\tilde{\mathbf{x}}$ is defined such that there is a mapping from $(\tilde{\mathbf{x}}, \mathbf{y})$ to $\mathbf{x}$ and that the mutual information $I^*_{\tilde{\mathbf{x}}, \mathbf{y}|\mathbf{z}} := \mathbb{E}_{p_{\text{model}}(\tilde{\mathbf{x}}, \mathbf{y}, \mathbf{z}; \theta^*)} \left[ \log \frac{p_{\text{model}}(\tilde{\mathbf{x}}, \mathbf{y}|\mathbf{z}; \theta^*)}{p_{\text{model}}(\tilde{\mathbf{x}}|\mathbf{z}; \theta^*) p_{\text{model}}(\mathbf{y}|\mathbf{z}; \theta^*)} \right]$ is zero (see discussion below). Then, given a sufficiently expressive partial encoder,*

$$\max_{\psi} \mathcal{O}_{\text{for}}(\theta^*, \phi^*, \psi) = \max_{\theta, \phi, \psi} \mathcal{O}_{\text{for}}(\theta, \phi, \psi). \quad (9)$$

This is proven in Appendix D and implies that we can use values of $\theta$ and $\phi$ learned using the unconditional VAE objective. Then to train a conditional generative model we need only optimise $\psi$. This leads to faster convergence, as well as faster training iterations since we only need to compute gradients for, and perform update steps on, the partial encoder's parameters $\psi$. For all of our experiments in Section 4 we use pretrained models released by Child (2020), leveraging between 2 GPU-weeks and 1 GPU-year of unconditional VAE training for each dataset. We name our method IPA (Inference in a Pretrained Artifact).

Table 1: Image completion results. Best performance is shown in **bold**, and second best is underlined. In the last row, $t$ denotes the "temperature" parameter (Child, 2020).

| Method | CIFAR-10 | | | ImageNet-64 | | | FFHQ-256 | | |
|---|---|---|---|---|---|---|---|---|---|
| | FID↓ | P-IDS↑ | LPIPS-GT↓ | FID↓ | P-IDS↑ | LPIPS-GT↓ | FID↓ | P-IDS↑ | LPIPS-GT↓ |
| ANP | 30.03 | 5.86 | .0447 | - | - | - | 39.95 | 0.93 | .256 |
| CE | 21.92 | 4.77 | .0628 | - | - | - | 39.02 | 0.66 | .267 |
| RFR | 44.35 | 2.76 | .0883 | - | - | - | 72.50 | 0.46 | .271 |
| PIC | 14.73 | 5.95 | .0332 | 40.0 | 0.24 | .170 | 11.60 | 2.76 | .169 |
| CoModGAN | 9.65 | 11.59 | .0326 | 20.2 | 7.09 | .160 | **2.33** | **13.57** | .143 |
| IPA-R | 19.21 | 8.56 | .0330 | 28.8 | 6.46 | .166 | 8.82 | 4.56 | .142 |
| IPA (ours) | 10.50 | 13.24 | **.0262** | 18.9 | 9.20 | .138 | 3.93 | 7.79 | .123 |
| IPA$_{t=0.85}$ (ours) | **8.61** | **14.19** | .0263 | **15.1** | **11.26** | **.133** | 3.29 | 8.50 | **.117** |

Theorem 3.2 relies on the assumption that the mutual information $I^*_{\tilde{\mathbf{x}},\mathbf{y}|\mathbf{z}}$ is zero; as we argue in Appendix D, this is true for inpainting and also "approximately" holds if $\mathbf{y}$ consists of high-level features. When lower level features are conditioned on, e.g. for super-resolution, there may be a significant gap between the left- and right- hand sides of Eq. (9). Theorem 3.2 also applies only if the unconditional VAE parameters are learned on the same dataset as the conditional VAE is trained on; otherwise there will be a mismatch between the form of $p_{\text{data}}$ used in Eq. (1) to fit $\theta^*$ and $\phi^*$, and the form of $p_{\text{data}}$ implicit in the $\mathcal{O}_{\text{for}}$ objective. However we find empirically that we can use unconditional VAE parameters trained on ImageNet (Deng et al., 2009) with IPA on several other photo datasets.

## 4 EXPERIMENTS

**Comparison to image completion baselines** We create an IPA image completion model based on the VD-VAE unconditional architecture (Child, 2020), and evaluate it for image completion on three datasets: CIFAR-10 (Krizhevsky et al., 2009), ImageNet-64 (Deng et al., 2009), and FFHQ-256 (Karras et al., 2019). We compare against four baselines: Co-Modulated Generative Adversarial Networks (CoModGAN) (Zhao et al., 2021); Pluralistic Image Completion (PIC) (Zheng et al., 2019); Context Encoders (CE) (Pathak et al., 2016); and Attentive Neural Processes (ANP) (Kim et al., 2019). We also considered two further methods: we show samples from VQ-VAE (Peng et al., 2021) (but not quantitative results which were too slow to compute because it takes roughly one minute per test image); and we report results for Recurrent Feature Reasoning for Image Inpainting (RFR) (Li et al., 2020a) with the caveat that it is slow to run on images with many missing pixels and so, although it used a similar computational budget to the other models, its training did not converge.

Given pretrained unconditional VAE parameters, IPA is faster to train than the best-performing baseline, CoModGAN. IPA takes 115 GPU-hours to train on CIFAR-10, and under 7 GPU-weeks on FFHQ-256. The CoModGAN models are trained for 270 GPU-hours and 8 GPU-weeks respectively. We provide more training details in Appendix G.

We report the FID (Heusel et al., 2017) and P-IDS (Zhao et al., 2021) metrics between a set of sampled completions from each method and a reference set. Broadly speaking, these measure the sample quality. To investigate how well the samples capture all modes of $p_{\text{data}}(\mathbf{x}|\mathbf{y})$, we also report the LPIPS-GT. We compute this using LPIPS (Zhang et al., 2018), a measure of distance between two images. Specifically, we compute the average over test pairs $(\mathbf{x}, \mathbf{y})$ of $\min_{k=1}^{K}(\text{LPIPS}(\mathbf{x}^{(k)}, \mathbf{x}))$, with each $\mathbf{x}^{(k)} \sim p_{\text{cond}}(\cdot|\mathbf{y})$. As $K \to \infty$, the LPIPS-GT should tend to zero if the ground truth is always within the support of $p_{\text{cond}}(\mathbf{x}|\mathbf{y})$. If not, the LPIPS-GT will remain high, penalising methods which miss modes of the posterior. We use $K = 100$. We confirm in Appendix H that the LPIPS-GT correlates with diversity metrics used by related work (Zhu et al., 2017; Li et al., 2020b).

For the image completion tasks, we sample from $p_{\text{data}}(\mathbf{x}, \mathbf{y})$ by first sampling an image $\mathbf{x}$ from the dataset, and then sampling an image-sized binary mask $m$ from the freeform mask distribution used by Zhao et al. (2021), which is itself based on Yu et al. (2018). We then set $\mathbf{y} = \text{concatenate}(\mathbf{x} \odot m, m)$. Here, $\odot$ is a pixel-wise multiplication operation which removes information from the missing pixels. The concatenation is performed along the channel dimension and makes it possible to distinguish between unobserved pixels and zero-valued pixels.

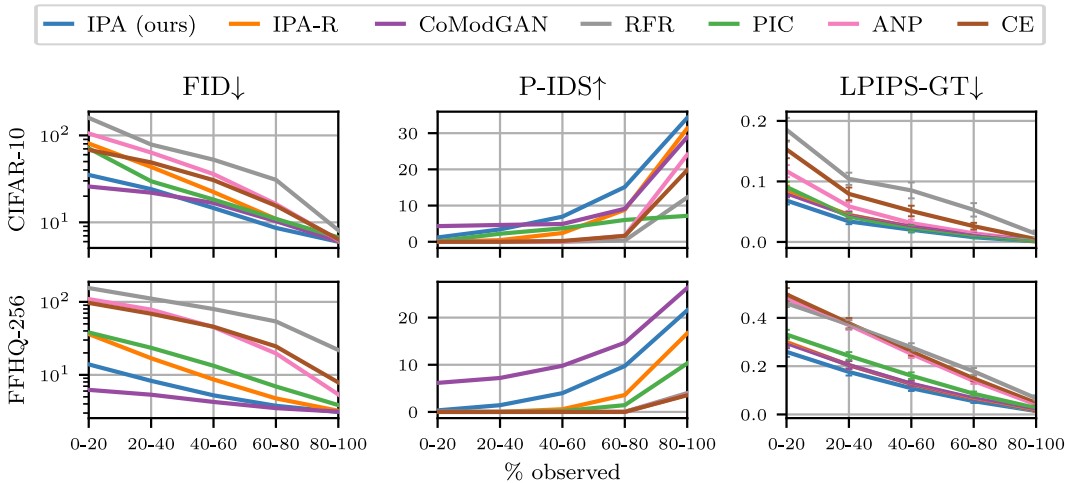

Figure 3: Test metrics on CIFAR-10 and FFHQ-256, plotted as a function of the mask distribution. Error bars on LPIPS-GT show the standard error of our estimate for a single trained network.

For evaluation, since the number of observed pixels in freeform masks varies considerably, we follow Zhao et al. (2021) and partition the mask distribution by conditioning the procedure to return a mask with the proportion of pixels observed within some range (0-20%, 20-40%, and so on) and report metrics for each range separately in Fig. 3 (or Appendix H for ImageNet-64). To summarise the overall performance in Table 1, we sample masks from a uniformly-weighted mixture distribution over these five partitions. In terms of the LPIPS-GT scores in Table 1, IPA outperforms the best baselines by roughly 20%. Figure 3 shows that there is an improvement for any proportion of observed pixels. This suggests that IPA produces reliably diverse samples with good coverage of $p_{\text{data}}(\mathbf{x}|\mathbf{y})$. In contrast, we believe that the GAN-based approaches occasionally miss modes of $p_{\text{data}}(\mathbf{x}|\mathbf{y})$ and can therefore fail to capture the ground-truth. This hypothesis is supported by samples from CoModGAN we display in Appendix K. In terms of sample fidelity, as measured by both FID and P-IDS, IPA outperforms all baselines on CIFAR-10 when over $40\%$ of the image is observed, and on ImageNet-64 when over $20\%$ is observed. IPA comes second to CoModGAN when less is observed and on FFHQ-256.

**Edges-to-photos**  We provide an additional demonstration of IPA on the Edges2Shoes and Edges2Handbags datasets (Isola et al., 2016), where the task is to generate an image conditioned on the output of an edge detector applied to that image. We downsample the datasets to $64 \times 64$ so that we can use unconditional VAEs pretrained on ImageNet (Deng et al., 2009) at this resolution by Child (2020). We show in Fig. 4 that IPA is useful for these tasks, and provide further discussion below. The images generated are diverse and photorealistic, as shown in Appendix L.

**Effectiveness of pretraining**  We now seek to determine how important the pretrained unconditional VAE weights are to IPA. We compare IPA with conditional VAEs which use IPA's architecture but are trained from scratch, which we will refer to as "from-scratch" baselines. That is, $\theta$ and $\phi$ are randomly initialised and trained to maximise Eq. (7) (with $\tilde{\mathbf{x}} := \mathbf{x}$) along with $\psi$. With an infinite training budget, the end-to-end training of the from-scratch baselines is likely to lead them to outperform any IPA models. Nevertheless it is apparent from Fig. 4 that, in the more realistic situation of a finite training budget, using IPA can be beneficial. This is the case even for training budgets of up to a few GPU-weeks on the relatively small CIFAR-10 dataset. In fact, even with only a couple of days of training, IPA on CIFAR-10 (with CIFAR-10 pretraining) achieves better FID and ELBO scores than the from-scratch baseline trained for several weeks. For ImageNet-64, IPA performs better after 4 hours than the from-scratch baseline does after a week. For Edges2Handbags and Edges2Shoes, training with IPA for 2 days yields performance similar to or better than training with the from-scratch baseline for 1 week, as measured by the ELBO. This is despite IPA on these datasets using a trained ImageNet-64 model rather than a model pretrained on those specific datasets, supporting our suggestion that the dataset used for pretraining need not exactly match what IPA is then trained on. Measured by the FID score, IPA's performance is even more appealing: wherever ELBOs are similar between IPA and the from-scratch baselines, IPA achieves a significantly better

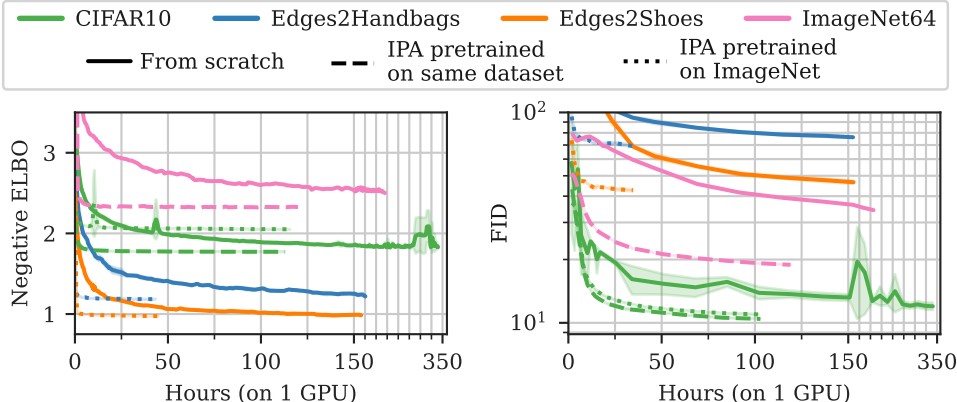

Figure 4: ELBO (Eq. (7) computed with $\tilde{\mathbf{x}} := \mathbf{x}$) and FID during training using IPA with pretraining on the same dataset, IPA with pretraining on ImageNet, and when trained from scratch. Error bars show standard deviations computed with 3 runs. IPA makes training faster and lower-variance.

FID score. We see that IPA pretrained on ImageNet is less effective for CIFAR-10 than it is for the edges-to-photos datasets, but it improves on the from-scratch baseline in terms of ELBO for the first 30 hours of training, and in terms of FID until the from-scratch baseline is trained for several weeks.

**An alternative training objective** In Table 1 and Fig. 3, we report results for IPA-R, a variation of IPA with a different training objective corresponding to a mode-seeking KL divergence. IPA almost always outperforms IPA-R, but we nonetheless provide a full description of IPA-R in Appendix E.

## 5 INPAINTING FOR BAYESIAN EXPERIMENTAL DESIGN

In this section, we explore a potential application for stochastic image completion that requires a faithful representation of the posterior $p_{\text{data}}(\mathbf{x}|\mathbf{y})$. In particular, we consider whether it is possible to automatically target a chest x-ray at areas most likely to reveal abnormalities. This could avoid the need to scan the entire chest and so bring benefits including reducing the patient's radiation exposure. While doing so is not possible with standard x-ray machines (which do not take multiple scans consecutively), and would need to be extensively validated before use in a clinical setting, we believe this is a worthwhile avenue to explore. Specifically, our imagined system performs a series of x-ray scans, each targeted at only a small portion of the area of interest. We can select the coordinates $l_t = (x_t, y_t)$ of the location to scan at each step $t$, and this selection can be informed by what was observed in the previous scans. The task we consider is how to select $l_t$ to be maximally informative. In particular, assume we wish to infer a variable $v$ representing, e.g., whether the patient has a particular illness. Bayesian optimal experimental design (BOED) (Chaloner & Verdinelli, 1995) provides a framework to select a value of $l_t$ that is maximally informative about $v$. It involves taking a Bayesian perspective on the problem of estimating $v$. We have one posterior distribution over $v$ after taking scans at $l_1, \ldots, l_{t-1}$ and another (typically lower entropy) distribution after conditioning on a scan at $l_t$ as well. The *expected information gain*, or EIG, quantifies the utility of the choice of $l_t$ as the expected difference in entropy between these two distributions. Using BOED involves estimating the EIG and selecting the scan location, $l_t$, to minimise it.

We use an estimator for the EIG similar to that of Harvey et al. (2019). It requires two components: **(I)** A neural network trained to classify $v$ given a series of scans at locations $l_1, \ldots, l_t$. This outputs a classification distribution which we denote $g(v|f_{l_1,\ldots,l_t}(\mathbf{x}))$, where $f_{l_1,\ldots,l_t}$ is a function mapping from an image to the values of the pixels observed by scans at $l_1, \ldots, l_t$. We use this classification distribution as an approximation of the posterior over $v$, whose entropy we attempt to minimise by performing BOED. **(II)** A method for sampling image completions conditioned on some observed pixel values $f_{l_1,\ldots,l_{t-1}}(\mathbf{x})$. Harvey et al. (2019) used a "stochastic image completion" module which contributed significant complexity to their method. We entirely replace this with IPA.

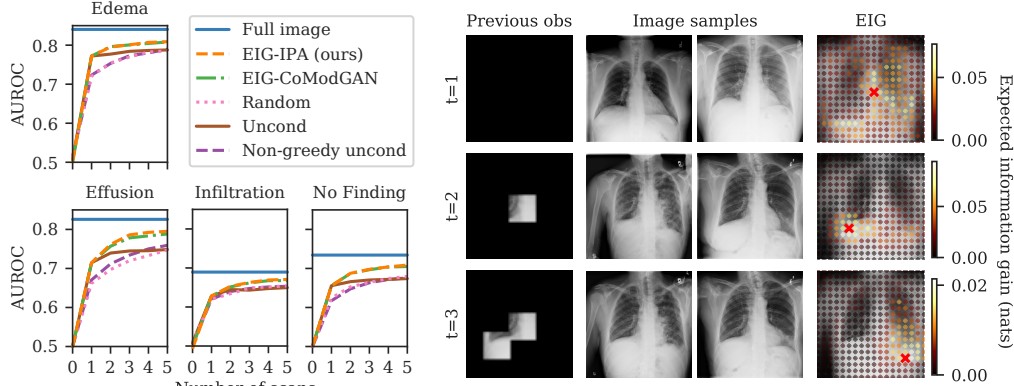

Figure 5: **Left:** Classification AUROC scores after $1, \ldots, 5$ scans chosen with each method. Scores for the "EIG-" methods more quickly approach the upper bound achieved by processing the full image. **Right:** Visualisation of BOED used to select three scan locations for diagnosing 'Effusion'. The left column shows the observations made prior to each time step. We then show samples from IPA (or the dataset when $t = 1$). The rightmost column shows the EIG overlaid on the pixel-space average of sampled images, with the optimal $l_t$ marked by a red cross.

Let the pixel values observed so far be $\mathbf{y}_{l_1, \ldots, l_{t-1}} = f_{l_1, \ldots, l_{t-1}}(\mathbf{x})$ for a latent image $\mathbf{x}$. Given these, we estimate the EIG of location $l_t$ as

$$\mathrm{EIG}(l_t; \mathbf{y}_{l_1, \ldots, l_{t-1}}) \approx \overbrace{\mathcal{H}\left[\frac{1}{N}\sum_{n=1}^{N} g(\cdot | f_{l_1, \ldots, l_t}(\mathbf{x}^{(n)}))\right]}^{\text{entropy after } t-1 \text{ scans}} - \overbrace{\frac{1}{N}\sum_{n=1}^{N}\mathcal{H}\left[g(\cdot | f_{l_1, \ldots, l_t}(\mathbf{x}^{(n)}))\right]}^{\text{expected entropy after } t \text{ scans}}, \quad (10)$$

where $\mathbf{x}^{(1)}, \ldots, \mathbf{x}^{(N)}$ are sampled image completions from IPA given $\mathbf{y}_{l_1, \ldots, l_{t-1}}$. In Appendix I we report hyperparameters, provide further details of our EIG estimator, and compare it to the estimators used in related work. To select $l_t$, we simply estimate $\mathrm{EIG}(l_t; \mathbf{y}_{l_1, \ldots, l_{t-1}})$ for many different values of $l_t$ and select the value which maximises it. This process of selecting $l_t$ and then taking a scan is repeated for each $t = 1, \ldots, T$.

We experiment on the NIH Chest X-ray 14 dataset (Wang et al., 2017) at $256 \times 256$ resolution. We simulate a scanner which returns a $64 \times 64$ pixel patch from this image, and the task is to diagnose the binary presence or absence of an illness. We run separate experiments diagnosing each of edema, effusion, infiltration and "no finding" (an additional label meaning there are no diagnosed illnesses). With appropriate data, this framework could be extended to also infer the severity of a given illness. We envisage BOED being used to select scan locations for an x-ray without necessarily performing an automated diagnosis. However, to quantify the informativeness of the chosen locations, Fig. 5 shows the results of using $g$ to perform a diagnosis, or classification, based on the chosen scan locations. Since the conditional distribution $g$ (used to estimate the EIG) depends on which illness we are classifying, the choice of scan locations is different in each case. We compare against a baseline where the image completion is performed by CoModGAN (our best-performing image completion baseline) rather than IPA, as well as numerous baselines which choose scan locations without image completion; see Appendix I for details.

Our method (denoted EIG-IPA) narrowly but consistently outperforms EIG-CoModGAN. We hypothesise that this is due to the aforementioned tendency of CoModGAN to sometimes collapse to a single mode of the posterior, and exhibit an example of this behaviour on the x-ray dataset in Appendix L. In the BOED context, such "overconfident" image completion could lead to salient scan locations being ignored. Nonetheless, both EIG-IPA and EIG-CoModGAN significantly outperform the other baselines, giving performance much closer to the upper bound of a CNN with access to the entire image. Another benefit of the "EIG-" approaches is that the choice of scan locations is highly interpretable; we can see why a particular location was chosen with visualisations similar to the right of Fig. 5. This shows the sampled images $\mathbf{x}^{(n)}$ and the estimated EIG for each $l_t$. In Appendix I, we show that we can further quantify the contribution of each $\mathbf{x}^{(n)}$ to the estimated EIG for each $l_t$.

## 6 RELATED WORK

**Inference in pretrained VAEs** Several prior studies perform conditional generation using a previously trained unconditional VAE. Like us, Rezende et al. (2014); Nguyen et al. (2016); Wu et al. (2018) do so through inference in the VAE's latent space. However, they use non-amortized inference (Gibbs sampling, variational inference, and MCMC respectively), leading to slow sampling times for any new $\mathbf{y}$. Duan et al. (2019) learn variational distributions over $\mathbf{z}$ for every possible value of $\mathbf{y}$, but this is not possible when $\mathbf{y}$ is high-dimensional or continuous-valued. Yeh et al. (2017) fit the latent variables of a GAN given observations, but this is neither amortized nor probabilistic.

**Conditional VAEs** Past research on conditional VAEs (Sohn et al., 2015; Zheng et al., 2019; Ivanov et al., 2018; Wan et al., 2021) has generally been unable to take advantage of pretrained weights as we have due to a difference in architectures: unlike almost all prior work, the IPA decoder does not receive $\mathbf{y}$ as input. The dependence between $\mathbf{y}$ and the decoder's output must therefore be expressed solely through the conditional distribution over the latent variables, $c(\mathbf{z}|\mathbf{y})$. This is a crucial difference because it means that the decoder can have exactly the same architecture as that of an unconditional VAE. This is key to letting us copy the pretrained weights of an unconditional VAE to speed up training. The exception to the above is Ma et al. (2018) who, like us, use a conditional VAE decoder with no dependence on $\mathbf{y}$. Their training objective and use case are different, however, and they do not consider using pretrained models or use an architecture which can scale to photorealistic images. Leveraging unconditional VAEs lets us drastically reduce the computational budget required to train a conditional VAE. We believe that this paper is the first to demonstrate photorealistic image completion with conditional VAEs at resolutions as high as $256 \times 256$. Another benefit of the decoder having no dependence on $\mathbf{y}$ is that it makes impossible the "posterior collapse" phenomenon discussed by Zheng et al. (2019), in which a conditional VAE's decoder learns to ignore $\mathbf{z}$ and produce outputs conditioned solely on $\mathbf{y}$.

**Image completion** Early work on image completion, both before (Bertalmio et al., 2000; 2001; Ballester et al., 2001; Levin et al., 2003; Criminisi et al., 2003) and after (Köhler et al., 2014; Ren et al., 2015) deep learning became the dominant approach, aimed to deterministically fill in missing pixels in images. Even many methods incorporating generative adversarial networks (GANs), which were introduced by Goodfellow et al. (2014) as a tool to learn distributions, have been found to result in little or no diversity in the completions produced for a given input (Song et al., 2018; Yu et al., 2018; 2019; Pathak et al., 2016; Iizuka et al., 2017). However, some recent methods have managed to obtain diverse completions using the GAN framework (Zhao et al., 2020; 2021; Liu et al., 2021). Another approach is to sample low-resolution images using VAEs (Zheng et al., 2019; Peng et al., 2021) or transformer-based sequence models (Zheng et al., 2021; Wan et al., 2021), and then use a GAN for upsampling. In contrast, we use a VAE to model image completions at the full resolution. As well as ensuring diverse coverage of the posterior, using such a likelihood-based model enables applications such as out-of-distribution detection for inputs $\mathbf{y}$, which we demonstrate in Appendix J. Another related approach is that of Song et al. (2020), who present a stochastic differential equation-based image model. This can be used for image completion but sampling is slow.

## 7 DISCUSSION AND CONCLUSION

We have presented IPA, a method to adapt an unconditional VAE into a conditional model. Image completions generated with IPA are close to the state-of-the-art in terms of visual fidelity, and improve on all baselines in terms of their coverage of the posterior as measured by LPIPS-GT. This high-fidelity coverage of the posterior makes IPA ideal for use in Bayesian optimal experimental design, as demonstrated. Our theoretical results suggest that, for the applications presented, using the weights of an unconditional VAE is approximately as good as training a conditional VAE from scratch. We note however, that there are applications for which these results will not hold (e.g. superresolution). They also provide no guarantees when pretraining is performed on a different dataset, although we show empirically that IPA can still be effective in this case. Future work may look more rigorously at these settings or further improve the image samples by, e.g., using a partial encoder with more expressive distributions. Preliminary experiments using normalizing flows helped improve the ELBO, but with little impact on the FID scores.

ETHICS STATEMENT

Our proposed method obtains more complete coverage of a conditional image distribution than our baselines. Since we so not directly push the state-of-the-art in terms of realism (and thereby aid, e.g., the creation of "deepfakes"), we are not aware of any obviously nefarious applications which will benefit. A potential positive impact is that, by providing better coverage of the posterior distribution, our work leads to better representation of minority groups in certain applications.

For a real-world application of the possible medical imaging procedure outlined in Section 5, we note that various challenges exist. In addition to the development of appropriate x-ray imaging hardware, these include more thoroughly investigating potential sources of failure "in the wild", and mitigating dataset biases. Working on the latter in particular is important to ensure that any decisions made are not worse for groups that are underrepresented in the dataset. Mitigating this would require more careful dataset curation, and possibly a new source of data.

ACKNOWLEDGMENTS

We acknowledge the support of the Natural Sciences and Engineering Research Council of Canada (NSERC), the Canada CIFAR AI Chairs Program, and the Intel Parallel Computing Centers program. Additional support was provided by UBC's Composites Research Network (CRN), and Data Science Institute (DSI). This research was enabled in part by technical support and computational resources provided by WestGrid (www.westgrid.ca), Compute Canada (www.computecanada.ca), and Advanced Research Computing at the University of British Columbia (arc.ubc.ca). WH acknowledges support by the University of British Columbia's Four Year Doctoral Fellowship (4YF) program.

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

Table 2: Definitions for symbols used.

| Symbol | Definition |
|---|---|
| $\mathbf{z}$ | VAE's latent variables. |
| $\mathbf{x}$ | Data we learn a generative model of. |
| $\mathbf{y}$ | Data on which our generative model is conditioned. |
| $\tilde{\mathbf{x}}$ | The part of $\mathbf{x}$ which the conditional VAE models, defined as all pixels not in $\mathbf{y}$ for image completion, or simply $\tilde{\mathbf{x}} := \mathbf{x}$ for other conditional generation tasks. |
| $\theta$ | Parameters for VAE's prior and decoder. |
| $\phi$ | Parameters for VAE's encoder. |
| $p_{\text{data}}(\mathbf{x}, \mathbf{y})$ | Distribution from which we assume training/test instances are i.i.d. samples. |
| $p_{\text{model}}(\mathbf{z})$ | Learned prior of VAE. Implicitly parameterised by $\theta$ when not otherwise specified. |
| $p_{\text{model}}(\mathbf{x}|\mathbf{z})$ | Distribution output by VAE's decoder given $\mathbf{z}$. Implicitly parameterised by $\theta$ when not otherwise specified. |
| $q(\mathbf{z}|\mathbf{x})$ | Distribution output by VAE's encoder given $\mathbf{x}$. Implicitly parameterised by $\phi$ when not otherwise specified. |
| $p_{\text{model}}(\mathbf{z}, \mathbf{x}, \mathbf{y})$ | Joint distribution defined as $p_{\text{model}}(\mathbf{z})p_{\text{model}}(\mathbf{x}|\mathbf{z})p_{\text{data}}(\mathbf{y}|\mathbf{x})$. |
| $r(\mathbf{z}, \mathbf{x}, \mathbf{y})$ | Joint distribution defined as $p_{\text{data}}(\mathbf{x}, \mathbf{y})q(\mathbf{z}|\mathbf{x})$. |
| $\psi$ | Partial encoder parameters. |
| $c(\mathbf{z}|\mathbf{y})$ | Distribution output by partial encoder. Implicitly parameterised by $\psi$. |
| $p_{\text{cond}}(\mathbf{x}|\mathbf{y})$ | Distribution over from which images are sampled given a conditioning input. Defined as $\int p_{\text{model}}(\mathbf{x}|\mathbf{z})c(\mathbf{z}|\mathbf{y})\mathrm{d}\mathbf{z}$. |
| $p^*(\mathbf{z}, \mathbf{x}, \mathbf{y})$ | Distribution with parameters $\theta^*$ and $\phi^*$ that are optimal as defined in Appendix D. Defined equivalently as $p_{\text{model}}(\mathbf{z}, \mathbf{x}, \mathbf{y}; \theta^*)$ or $p_{\text{data}}(\mathbf{x}, \mathbf{y})q(\mathbf{z}|\mathbf{x}; \phi^*)$. |

Symbols for Bayesian optimal experimental design:

| | |
|---|---|
| $l_t$ | The location scanned at time $t$. |
| $v$ | Variable we wish to infer. |
| $\mathbf{y}_{l_1, \ldots, l_t}$ | Image in which all pixel values are missing except for those observed by scans at locations $l_1, \ldots, l_t$. |
| $f_{l_1, \ldots, l_t}$ | Function mapping from an image $\mathbf{x}$ to $\mathbf{y}_{l_1, \ldots, l_t}$. |
| $g(v|\mathbf{y})$ | Classification distribution for $v$ given partially observed image $\mathbf{y}$. |

## A  NOTATION

We provide Table 2 for reference as a concise list of definitions for each symbol used. More thorough explanations are provided where symbols are introduced in the text

## B  EXPANDED DERIVATIONS

### B.1  EXPANDED DERIVATION OF EQ. (2)

Here we expand on the steps to show that the expectation of the ELBO in Eq. (1) is equivalent to the lower bound on the negative entropy of $p_{\text{data}}(\mathbf{x})$ in Eq. (2).

$$\mathbb{E}_{p_{\text{data}}(\mathbf{x})}\left[\text{ELBO}(\theta, \phi, \mathbf{x})\right] = \mathbb{E}_{p_{\text{data}}(\mathbf{x})}\mathbb{E}_{q(\mathbf{z}|\mathbf{x};\phi)}\left[\log \frac{p_{\text{model}}(\mathbf{z};\theta)p_{\text{model}}(\mathbf{x}|\mathbf{z};\theta)}{q(\mathbf{z}|\mathbf{x};\phi)}\right] \tag{11}$$

$$= \mathbb{E}_{p_{\text{data}}(\mathbf{x})q(\mathbf{z}|\mathbf{x};\phi)}\left[\log \frac{p_{\text{model}}(\mathbf{z}, \mathbf{x};\theta)}{q(\mathbf{z}|\mathbf{x};\phi)}\right] \tag{12}$$

$$= \mathbb{E}_{p_{\text{data}}(\mathbf{x})q(\mathbf{z}|\mathbf{x};\phi)}\left[\log p_{\text{data}}(\mathbf{x}) + \log \frac{p_{\text{model}}(\mathbf{z}, \mathbf{x};\theta)}{p_{\text{data}}(\mathbf{x})q(\mathbf{z}|\mathbf{x};\phi)}\right] \tag{13}$$

$$= \mathbb{E}_{p_{\text{data}}(\mathbf{x})}\left[\log p_{\text{data}}(\mathbf{x})\right] + \mathbb{E}_{p_{\text{data}}(\mathbf{x})q(\mathbf{z}|\mathbf{x};\phi)}\left[\log \frac{p_{\text{model}}(\mathbf{z}, \mathbf{x};\theta)}{p_{\text{data}}(\mathbf{x})q(\mathbf{z}|\mathbf{x};\phi)}\right] \tag{14}$$

$$= -\mathcal{H}\left[p_{\text{data}}(\mathbf{x})\right] - \text{KL}\left(p_{\text{data}}(\mathbf{x})q(\mathbf{z}|\mathbf{x};\phi) \,\|\, p_{\text{model}}(\mathbf{z}, \mathbf{x};\theta)\right) \tag{15}$$

as written in Eq. (2).

## B.2 EXPANDED DERIVATION OF EQ. (7)

Here we prove Eq. (7), which states that IPA's training objective is a lower bound on $\mathbb{E}_{p_{\text{data}}(\mathbf{x},\mathbf{y})}[\log p_{\text{cond}}(\tilde{\mathbf{x}}|\mathbf{y})]$.

$$\mathcal{O}_{\text{for}}(\theta,\phi,\psi) = \mathbb{E}_{p_{\text{data}}(\mathbf{x},\mathbf{y})}\mathbb{E}_{q(\mathbf{z}|\mathbf{x})}\left[\log\frac{p_{\text{model}}(\tilde{\mathbf{x}}|\mathbf{z})c(\mathbf{z}|\mathbf{y})}{q(\mathbf{z}|\mathbf{x})}\right] \tag{16}$$

$$\leq \mathbb{E}_{p_{\text{data}}(\mathbf{x},\mathbf{y})}\left[\log\mathbb{E}_{q(\mathbf{z}|\mathbf{x})}\left[\frac{p_{\text{model}}(\tilde{\mathbf{x}}|\mathbf{z})c(\mathbf{z}|\mathbf{y})}{q(\mathbf{z}|\mathbf{x})}\right]\right] \tag{17}$$

$$= \mathbb{E}_{p_{\text{data}}(\mathbf{x},\mathbf{y})}\left[\log\mathbb{E}_{c(\mathbf{z}|\mathbf{y})}[p_{\text{model}}(\tilde{\mathbf{x}}|\mathbf{z})]\right] \tag{18}$$

$$= \mathbb{E}_{p_{\text{data}}(\mathbf{x},\mathbf{y})}\left[\log p_{\text{cond}}(\tilde{\mathbf{x}}|\mathbf{y})\right], \tag{19}$$

where the final step follows from the definition of $p_{\text{cond}}$ in Eq. (4).

## C PROOF AND DISCUSSION OF THEOREM 3.1

### C.1 PROOF

**Statement.** *For any set $\Psi$ of permissible values of $\psi$, and for any $\theta\in\Theta$ and $\phi\in\Phi$,*

$$\underset{\psi\in\Psi}{\arg\max}\,\mathcal{O}_{\text{for}}(\theta,\phi,\psi) = \underset{\psi\in\Psi}{\arg\min}\,\mathbb{E}_{p_{\text{data}}(\mathbf{y})}\left[KL\big(r(\mathbf{z}|\mathbf{y})\,\|\,c(\mathbf{z}|\mathbf{y})\big)\right]. \tag{8}$$

**Proof.** We can decompose $\mathcal{O}_{\text{for}}$ as follows. Starting by multiplying both sides of the fraction by the intractable conditional distribution $r(\mathbf{z}|\mathbf{y})$,

$$\mathcal{O}_{\text{for}}(\theta,\phi,\psi) = \mathbb{E}_{r(\mathbf{z},\mathbf{x},\mathbf{y})}\left[\log\frac{p_{\text{model}}(\tilde{\mathbf{x}}|\mathbf{z})c(\mathbf{z}|\mathbf{y})r(\mathbf{z}|\mathbf{y})}{q(\mathbf{z}|\mathbf{x})r(\mathbf{z}|\mathbf{y})}\right] \tag{20}$$

$$= \mathbb{E}_{r(\mathbf{z},\mathbf{x},\mathbf{y})}\left[\log\frac{p_{\text{model}}(\tilde{\mathbf{x}}|\mathbf{z})r(\mathbf{z}|\mathbf{y})}{q(\mathbf{z}|\mathbf{x})}\right] - \mathbb{E}_{r(\mathbf{z},\mathbf{y})}\left[\log\frac{r(\mathbf{z}|\mathbf{y})}{c(\mathbf{z}|\mathbf{y})}\right] \tag{21}$$

$$= C(\theta,\phi) - \mathbb{E}_{p_{\text{data}}(\mathbf{y})}\left[\text{KL}\big(r(\mathbf{z}|\mathbf{y})\,\|\,c(\mathbf{z}|\mathbf{y})\big)\right] \tag{22}$$

where $C(\theta,\phi)$ is a grouping of terms which do not depend on the partial encoder's parameters $\psi$. Any values of $\psi$ which maximise $\mathcal{O}_{\text{for}}(\theta,\phi,\psi)$ will therefore also minimise $\mathbb{E}_{p_{\text{data}}(\mathbf{y})}\left[\text{KL}\big(r(\mathbf{z}|\mathbf{y})\,\|\,c(\mathbf{z}|\mathbf{y})\big)\right]$ for the same values of $\theta$ and $\phi$, which proves the theorem.

### C.2 BOUND ON $\tilde{\mathbf{x}}$-SPACE DIVERGENCE

The mass-covering KL divergence in Theorem 3.1 indicates that samples $\mathbf{z}\sim c(\cdot|\mathbf{y})$ will be diverse (when $r(\mathbf{z}|\mathbf{y})$ is diverse/uncertain). One would expect that samples of $\tilde{\mathbf{x}}$ conditioned on these diverse samples of $\mathbf{z}$ will also be diverse. Here we make a more formal argument by showing that minimising $\mathcal{O}_{\text{for}}$ corresponds to miminising an upper-bound on an expected mass-covering KL divergence in $\tilde{\mathbf{x}}$-space. Starting from the result in Appendix B.2,

$$\mathcal{O}_{\text{for}}(\theta,\phi,\psi) \leq \mathbb{E}_{p_{\text{data}}(\mathbf{x},\mathbf{y})}\left[\log p_{\text{cond}}(\tilde{\mathbf{x}}|\mathbf{y})\right] \tag{23}$$

$$= \mathbb{E}_{p_{\text{data}}(\mathbf{x},\mathbf{y})}\left[\log p_{\text{data}}(\tilde{\mathbf{x}}|\mathbf{y})\right] - \mathbb{E}_{p_{\text{data}}(\mathbf{y})}\left[\text{KL}(p_{\text{data}}(\tilde{\mathbf{x}}|\mathbf{y})\,\|\,p_{\text{cond}}(\tilde{\mathbf{x}}|\mathbf{y}))\right]. \tag{24}$$

Rearranging this gives

$$\mathbb{E}_{p_{\text{data}}(\mathbf{y})}\left[\text{KL}(p_{\text{data}}(\tilde{\mathbf{x}}|\mathbf{y})\,\|\,p_{\text{cond}}(\tilde{\mathbf{x}}|\mathbf{y}))\right] \leq \mathbb{E}_{p_{\text{data}}(\mathbf{x},\mathbf{y})}\left[\log p_{\text{data}}(\tilde{\mathbf{x}}|\mathbf{y})\right] - \mathcal{O}_{\text{for}}(\theta,\phi,\psi). \tag{25}$$

Maximising $\mathcal{O}_{\text{for}}$ therefore minimises an upper-bound on an expected mass-covering KL divergence in $\tilde{\mathbf{x}}$-space, providing a further suggestion that $\tilde{\mathbf{x}}\sim p_{\text{cond}}(\cdot|\mathbf{y})$ will be diverse.

# D  PROOF AND DISCUSSION OF THEOREM 3.2

## D.1  PROOF

**Statement.** *Assume that we have a sufficiently expressive encoder and decoder that there exist parameters $\theta^* \in \Theta$ and $\phi^* \in \Phi$ which make the unconditional VAE objective (Eq. (1)) equal to its upper bound of $-\mathcal{H}\left[p_{\mathrm{data}}(\mathbf{x})\right]$. Assume also that the mutual information $I^*_{\tilde{\mathbf{x}},\mathbf{y}|\mathbf{z}} := \mathbb{E}_{p_{\mathrm{model}}(\tilde{\mathbf{x}},\mathbf{y},\mathbf{z};\theta^*)}\left[\log \frac{p_{\mathrm{model}}(\tilde{\mathbf{x}},\mathbf{y}|\mathbf{z};\theta^*)}{p_{\mathrm{model}}(\tilde{\mathbf{x}}|\mathbf{z};\theta^*)p_{\mathrm{model}}(\mathbf{y}|\mathbf{z};\theta^*)}\right] = \mathbb{E}_{p_{\mathrm{model}}(\tilde{\mathbf{x}},\mathbf{y},\mathbf{z};\theta^*)}\left[\log \frac{p_{\mathrm{model}}(\tilde{\mathbf{x}}|\mathbf{z},\mathbf{y};\theta^*)}{p_{\mathrm{model}}(\tilde{\mathbf{x}}|\mathbf{z};\theta^*)}\right]$ is zero (meaning that $\tilde{\mathbf{x}}$ and $\mathbf{y}$ are conditionally independent given $\mathbf{z}$) and $\tilde{\mathbf{x}}$ is defined such that there exists a mapping from $(\tilde{\mathbf{x}}, \mathbf{y})$ to $\mathbf{x}$. Then, given a sufficiently expressive partial encoder $c(\mathbf{z}|\mathbf{y};\psi)$,*

$$\max_{\psi} \mathcal{O}_{\mathrm{for}}(\theta^*, \phi^*, \psi) = \max_{\theta,\phi,\psi} \mathcal{O}_{\mathrm{for}}(\theta, \phi, \psi). \tag{9}$$

**Proof.** We have used, and will continue to use, $p_{\mathrm{model}}(\cdot)$ and $q(\cdot)$ to refer to joint distributions parameterised by $\theta$ and $\phi$ respectively. Since $\theta^*$ and $\phi^*$ are defined as the maximisers of the unconditional ELBO in Eq. (2), and given our assumption of sufficient expressivity, we have $p_{\mathrm{model}}(\mathbf{z}, \mathbf{x}; \theta^*) = p_{\mathrm{data}}(\mathbf{x})q(\mathbf{z}|\mathbf{x};\phi^*)$. We can therefore introduce notation for the equal joint distributions: $p^*(\mathbf{z}, \mathbf{x}, \mathbf{y}) = p_{\mathrm{model}}(\mathbf{z}, \mathbf{x}, \mathbf{y}; \theta^*) = p_{\mathrm{data}}(\mathbf{x}, \mathbf{y})q(\mathbf{z}|\mathbf{x};\phi^*)$. We note that we can also write the joint distribution $p^*(\mathbf{z}, \mathbf{x}, \tilde{\mathbf{x}}, \mathbf{y}) = p_{\mathrm{data}}(\mathbf{x}, \mathbf{y})q(\mathbf{z}|\mathbf{x};\phi^*)p(\tilde{\mathbf{x}}|\mathbf{x}, \mathbf{y})$, where $p(\tilde{\mathbf{x}}|\mathbf{x}, \mathbf{y}) = \delta_{g(\mathbf{x},\mathbf{y})}(\tilde{\mathbf{x}})$ is the Dirac distribution mapping $\mathbf{x}$ and $\mathbf{y}$ deterministically to $\tilde{\mathbf{x}} = g(\mathbf{x}, \mathbf{y})$. Recalling that we define $\tilde{\mathbf{x}}$ as either equal to $\mathbf{x}$ or as the dimensions of $\mathbf{x}$ not observed in $\mathbf{y}$, such a deterministic $g$ will always exist.

We consider the decomposition of $\mathcal{O}_{\mathrm{for}}$ shown in Eq. (22) and remind the reader that, given a sufficiently expressive partial encoder, the maximisation over $\psi$ will always make the KL divergence zero. That is,

$$\max_{\psi} \mathcal{O}_{\mathrm{for}}(\theta, \phi, \psi) = C(\theta, \phi) = \mathbb{E}_{r(\mathbf{z},\mathbf{x},\mathbf{y})}\left[\log \frac{p_{\mathrm{model}}(\tilde{\mathbf{x}}|\mathbf{z})r(\mathbf{z}|\mathbf{y})}{q(\mathbf{z}|\mathbf{x})}\right] \tag{26}$$

for any $\theta$ and $\phi$. Using $\theta^*$ and $\phi^*$, and therefore using $p^*$ in place of $p_{\mathrm{model}}$, $r$ and $q$, we can expand the left-hand side of Eq. (9).

$$\max_{\psi} \mathcal{O}_{\mathrm{for}}(\theta^*, \phi^*, \psi) = \mathbb{E}_{p^*(\mathbf{z},\mathbf{x},\mathbf{y})}\left[\log \frac{p^*(\tilde{\mathbf{x}}|\mathbf{z})p^*(\mathbf{z}|\mathbf{y})}{p^*(\mathbf{z}|\mathbf{x})}\right] \tag{27}$$

$$= \mathbb{E}_{p^*(\mathbf{z},\mathbf{x},\mathbf{y})}\left[\log \frac{p^*(\tilde{\mathbf{x}}|\mathbf{z},\mathbf{y})p^*(\mathbf{z}|\mathbf{y})}{p^*(\mathbf{z}|\mathbf{x})}\right] - \mathbb{E}_{p^*(\mathbf{z},\tilde{\mathbf{x}},\mathbf{y})}\left[\log \frac{p^*(\tilde{\mathbf{x}}|\mathbf{z},\mathbf{y})}{p^*(\tilde{\mathbf{x}}|\mathbf{z})}\right] \tag{28}$$

$$= \mathbb{E}_{p^*(\mathbf{z},\mathbf{x},\mathbf{y})}\left[\log \frac{p^*(\tilde{\mathbf{x}}|\mathbf{z},\mathbf{y})p^*(\mathbf{z}|\mathbf{y})}{p^*(\mathbf{z}|\mathbf{x})}\right] - I^*_{\tilde{\mathbf{x}},\mathbf{y}|\mathbf{z}} \tag{29}$$

$$= \mathbb{E}_{p^*(\mathbf{z},\mathbf{x},\mathbf{y})}\left[\log \frac{p^*(\mathbf{z},\tilde{\mathbf{x}}|\mathbf{y})}{p^*(\mathbf{z}|\mathbf{x})}\right] - I^*_{\tilde{\mathbf{x}},\mathbf{y}|\mathbf{z}} \tag{30}$$

$$= \mathbb{E}_{p^*(\mathbf{z},\mathbf{x},\mathbf{y})}\left[\log \frac{p^*(\mathbf{z}|\tilde{\mathbf{x}},\mathbf{y})p^*(\tilde{\mathbf{x}}|\mathbf{y})}{p^*(\mathbf{z}|\mathbf{x})}\right] - I^*_{\tilde{\mathbf{x}},\mathbf{y}|\mathbf{z}}. \tag{31}$$

The factorisation $p^*(\mathbf{z}, \mathbf{x}, \tilde{\mathbf{x}}, \mathbf{y}) = p_{\mathrm{data}}(\mathbf{x}, \mathbf{y})q(\mathbf{z}|\mathbf{x};\phi^*)\delta(\tilde{\mathbf{x}}|\mathbf{x}, \mathbf{y})$ implies that we can express $p^*(\mathbf{z}|\tilde{\mathbf{x}}, \mathbf{y})$ with a marginalisation: $p^*(\mathbf{z}|\tilde{\mathbf{x}}, \mathbf{y}) = \int p^*(\mathbf{z}|\mathbf{x})p^*(\mathbf{x}|\tilde{\mathbf{x}}, \mathbf{y})\mathrm{d}\mathbf{x}$. Since we have assumed that there is a deterministic function $f$ mapping $(\tilde{\mathbf{x}}, \mathbf{y})$ to $\mathbf{x}$, $p^*(\mathbf{x}|\tilde{\mathbf{x}}, \mathbf{y})$ is a Dirac distribution and the marginalisation becomes $p^*(\mathbf{z}|\tilde{\mathbf{x}}, \mathbf{y}) = \int p^*(\mathbf{z}|\mathbf{x})\delta(\mathbf{x}|\tilde{\mathbf{x}}, \mathbf{y})\mathrm{d}\mathbf{x} = p^*(\mathbf{z}|\mathbf{x} = f(\tilde{\mathbf{x}}, \mathbf{y}))$. We therefore proceed, rewriting $p^*(\mathbf{z}|\tilde{\mathbf{x}}, \mathbf{y})$ as $p^*(\mathbf{z}|\mathbf{x})$:

$$\max_{\psi} \mathcal{O}_{\mathrm{for}}(\theta^*, \phi^*, \psi) = \mathbb{E}_{p^*(\mathbf{z},\mathbf{x},\mathbf{y})}\left[\log \frac{p^*(\mathbf{z}|\mathbf{x})p^*(\tilde{\mathbf{x}}|\mathbf{y})}{p^*(\mathbf{z}|\mathbf{x})}\right] - I^*_{\tilde{\mathbf{x}},\mathbf{y}|\mathbf{z}} \tag{32}$$

$$= \mathbb{E}_{p^*(\mathbf{x},\mathbf{y})}\left[\log p^*(\tilde{\mathbf{x}}|\mathbf{y})\right] - I^*_{\tilde{\mathbf{x}},\mathbf{y}|\mathbf{z}} \tag{33}$$

$$= \mathbb{E}_{p_{\mathrm{data}}(\mathbf{x},\mathbf{y})}\left[\log p_{\mathrm{data}}(\tilde{\mathbf{x}}|\mathbf{y})\right] - I^*_{\tilde{\mathbf{x}},\mathbf{y}|\mathbf{z}}. \tag{34}$$

Now that we have this expansion of the left-hand side of Eq. (9), we can derive a related upper-bound on the right-hand side as follows. For any $\theta$, any $\phi$, and any $\psi$, we have from Appendix B.2:

$$\mathcal{O}_{\text{for}}(\theta, \phi, \psi) \leq \mathbb{E}_{p_{\text{data}}(\mathbf{x}, \mathbf{y})} \left[ \log p_{\text{cond}}(\tilde{\mathbf{x}}|\mathbf{y}) \right] \tag{35}$$

$$= \mathbb{E}_{p_{\text{data}}(\mathbf{x}, \mathbf{y})} \left[ \log p_{\text{data}}(\tilde{\mathbf{x}}|\mathbf{y}) \right] - \mathbb{E}_{p_{\text{data}}(\mathbf{y})} \left[ \text{KL}(p_{\text{data}}(\tilde{\mathbf{x}}|\mathbf{y}) \, \| \, p_{\text{cond}}(\tilde{\mathbf{x}}|\mathbf{y})) \right] \tag{36}$$

$$\leq \mathbb{E}_{p_{\text{data}}(\mathbf{x}, \mathbf{y})} \left[ \log p_{\text{data}}(\tilde{\mathbf{x}}|\mathbf{y}) \right]. \tag{37}$$

This bound must also hold for the maximum over $\theta$, $\phi$, and $\psi$, and so is an upper-bound on the right-hand side of Eq. (9):

$$\max_{\theta, \phi, \psi} \mathcal{O}_{\text{for}}(\theta, \phi, \psi) \leq \mathbb{E}_{p_{\text{data}}(\mathbf{x}, \mathbf{y})} \left[ \log p_{\text{data}}(\tilde{\mathbf{x}}|\mathbf{y}) \right]. \tag{38}$$

By relating Eq. (34) and Eq. (38), we have:

$$\max_{\theta, \phi, \psi} \mathcal{O}_{\text{for}}(\theta, \phi, \psi) \leq \max_{\psi} \mathcal{O}_{\text{for}}(\theta^*, \phi^*, \psi) + I^*_{\tilde{\mathbf{x}}, \mathbf{y}|\mathbf{z}}. \tag{39}$$

We then point out that we can also obtain an inequality in the other direction: since the right-hand side of Eq. (9) is a maximisation of the left-hand side over $\theta$ and $\phi$, it must be greater than or equal to the left-hand side:

$$\max_{\psi} \mathcal{O}_{\text{for}}(\theta^*, \phi^*, \psi) \leq \max_{\theta, \phi, \psi} \mathcal{O}_{\text{for}}(\theta, \phi, \psi). \tag{40}$$

Combining Eq. (39) and Eq. (40), we have:

$$0 \leq \max_{\theta, \phi, \psi} \mathcal{O}_{\text{for}}(\theta, \phi, \psi) - \max_{\psi} \mathcal{O}_{\text{for}}(\theta^*, \phi^*, \psi) \leq I^*_{\tilde{\mathbf{x}}, \mathbf{y}|\mathbf{z}}. \tag{41}$$

We now have bounds in both directions on the difference between the right- and left-hand sides. To conclude the proof, we repeat the assumption that $I^*_{\tilde{\mathbf{x}}, \mathbf{y}|\mathbf{z}} = 0$. The upper- and lower- bounds of Eq. (41) are therefore both zero, so we have the final result that $\max_{\psi} \mathcal{O}_{\text{for}}(\theta^*, \phi^*, \psi) = \max_{\theta, \phi, \psi} \mathcal{O}_{\text{for}}(\theta, \phi, \psi)$.

## D.2   DISCUSSION OF THE ASSUMPTION THAT $I^*_{\tilde{\mathbf{x}}, \mathbf{y}|\mathbf{z}} = 0$

The above proof of Theorem 3.2 relies on the assumption that $I^*_{\tilde{\mathbf{x}}, \mathbf{y}|\mathbf{z}} = \mathbb{E}_{p^*(\tilde{\mathbf{x}}, \mathbf{y}, \mathbf{z})} \left[ \log \frac{p^*(\tilde{\mathbf{x}}|\mathbf{z}, \mathbf{y})}{p^*(\tilde{\mathbf{x}}|\mathbf{z})} \right] = 0$. That is, under the joint distribution $p^*(\mathbf{z}, \mathbf{x}, \mathbf{y})$, $\tilde{\mathbf{x}}$ and $\mathbf{y}$ should be conditionally independent given $\mathbf{z}$. We now argue that this is true for image completion, and "close to" satisfied when $\mathbf{y}$ consists of high-level image features. This means that Theorem 3.2 is informative for inpainting, our edges-to-photos experiments and, e.g., class-conditional generation but may not even approximately hold for, e.g., super-resolution or image colourisation.

**Image completion**   In the VAE architecture we use, the likelihood $p_{\text{model}}(\mathbf{x}|\mathbf{z})$ is pixel-wise independent. In other words, if we write $\mathbf{x}$ as a set of pixels $\{\mathbf{x}_1, \ldots, \mathbf{x}_N\}$, we can factorise $p_{\text{model}}(\mathbf{x}|\mathbf{z})$ as $\prod_{i=1}^{N} p_{\text{model}}(\mathbf{x}_i|\mathbf{z})$. This means that the conditional mutual information between any two disjoint subsets of $\mathbf{x}$ (conditioned on $\mathbf{z}$) will be zero. Given our definition of $\tilde{\mathbf{x}}$ for image completion as the the set of purely unobserved pixels, we have by construction that $\tilde{\mathbf{x}}$ and $\mathbf{y}$ are disjoint. Therefore, the assumption that $I^*_{\tilde{\mathbf{x}}, \mathbf{y}|\mathbf{z}} = 0$ is satisfied for image completion.

**When $I^*_{\tilde{\mathbf{x}}, \mathbf{y}|\mathbf{z}}$ is non-zero.**   When $I^*_{\tilde{\mathbf{x}}, \mathbf{y}|\mathbf{z}}$ is non-zero, we can still use it as an upper-bound on the difference between the left- and right-hand side of Eq. (9) using Eq. (41). Moreover, we posit that $I^*_{\tilde{\mathbf{x}}, \mathbf{y}|\mathbf{z}}$ will typically be small when $\mathbf{y}$ represents high-level features (e.g. the edges in Edges2Photos or a class label) and we have a pixel-wise independent likelihood. This is because any high-level structure cannot be modelled by the pixel-wise independent likelihood $p^*(\tilde{\mathbf{x}}|\mathbf{z})$ and so must be captured by $\mathbf{z}$. Intuitively, high-level features $\mathbf{y}$ are unlikely to be informative about any pixel-level variations if all image structure spanning multiple pixels ($\mathbf{z}$) is already known. This is equivalent to saying that $I^*_{\tilde{\mathbf{x}}, \mathbf{y}|\mathbf{z}}$, the mutual information between $\tilde{\mathbf{x}}$ and $\mathbf{y}$ conditioned on $\mathbf{z}$, is likely to be small.

# E  AN ALTERNATIVE TRAINING OBJECTIVE

In this section, we describe an alternative objective previously used by Ma et al. (2018). We find experimentally that its performance is typically worse than $\mathcal{O}_{\text{for}}$, but it has the advantage of only requiring data $\mathbf{y} \sim p_{\text{data}}(\cdot)$ during training, and not $\mathbf{x}$. This may allow training in settings such as outpainting (Sabini & Rusak, 2018). The objective can be interpreted as a lower bound on $\log p_{\text{model}}(\mathbf{y})$:

$$\mathcal{O}_{\text{rev}}(\theta, \phi, \psi) = \mathbb{E}_{p_{\text{data}}(\mathbf{y})} \mathbb{E}_{c(\mathbf{z}|\mathbf{y})} \left[ \log \frac{p_{\text{model}}(\mathbf{z}, \mathbf{y})}{c(\mathbf{z}|\mathbf{y})} \right] \leq \mathbb{E}_{p_{\text{data}}(\mathbf{y})} \left[ \log p_{\text{model}}(\mathbf{y}) \right]. \tag{42}$$

Computing this objective requires a computation graph similar to that shown in Fig. 2c. Note one caveat with this objective is that using it involves computing and differentiating $p_{\text{model}}(\mathbf{z}, \mathbf{y})$, which requires a tractable likelihood $p_{\text{data}}(\mathbf{y}|\mathbf{x})$ (based on the definition of $p_{\text{model}}$ in Table 2). This is possible in image completion, but not in the general case of conditional generation.

The following theorem says that learning $\psi$ to maximise this objective is equivalent to minimising another KL divergence.

**Theorem E.1.** *For any set $\Psi$ of permissible values of $\psi$, and for any $\theta \in \Theta$ and $\phi \in \Phi$,*

$$\underset{\psi \in \Psi}{\arg\max} \, \mathcal{O}_{rev}(\theta, \phi, \psi) = \underset{\psi \in \Psi}{\arg\min} \, \mathbb{E}_{p_{\text{data}}(\mathbf{y})} \left[ KL\big(c(\mathbf{z}|\mathbf{y}; \psi) \, \| \, p_{\text{model}}(\mathbf{z}|\mathbf{y}; \theta)\big) \right]. \tag{43}$$

**Proof.** We show this by decomposing the objective:

$$\mathcal{O}_{\text{rev}}(\theta, \phi, \psi) = \mathbb{E}_{p_{\text{data}}(\mathbf{y})c(\mathbf{z}|\mathbf{y})} \left[ \log \frac{p_{\text{model}}(\mathbf{z}, \mathbf{y})}{c(\mathbf{z}|\mathbf{y})} \right] \tag{44}$$

$$= \mathbb{E}_{p_{\text{data}}(\mathbf{y})c(\mathbf{z}|\mathbf{y})} \left[ \log p_{\text{model}}(\mathbf{y}) - \log \frac{c(\mathbf{z}|\mathbf{y})}{p_{\text{model}}(\mathbf{z}|\mathbf{y})} \right] \tag{45}$$

$$= \mathbb{E}_{p_{\text{data}}(\mathbf{y})} \left[ \log p_{\text{model}}(\mathbf{y}) - \text{KL}\big(c(\mathbf{z}|\mathbf{y}) \, \| \, p_{\text{model}}(\mathbf{z}|\mathbf{y})\big) \right] \tag{46}$$

$$= D(\theta) - \mathbb{E}_{p_{\text{data}}(\mathbf{y})} \left[ \text{KL}\big(c(\mathbf{z}|\mathbf{y}) \, \| \, p_{\text{model}}(\mathbf{z}|\mathbf{y})\big) \right] \tag{47}$$

where $D(\theta)$ is not dependent on $\psi$. Therefore, for any $\theta$ and $\phi$, any values of $\psi \in \Psi$ which maximise $\mathcal{O}_{\text{rev}}(\theta, \phi, \psi)$ will also minimise $\mathbb{E}_{p_{\text{data}}(\mathbf{y})} \left[ \text{KL}\big(c(\mathbf{z}|\mathbf{y}) \, \| \, p_{\text{model}}(\mathbf{z}|\mathbf{y})\big) \right]$, proving the theorem. ∎

Theorem E.1 implies that maximising $\mathcal{O}_{\text{rev}}$ will minimise a "reverse" KL divergence, causing $c(\mathbf{z}|\mathbf{y})$ to exhibit mode-seeking behaviour. We denote the method of training with this objective IPA-R (inference in a pretrained artifact with the reverse KL). As with IPA, we use pretrained $\theta$ and $\phi$ when training IPA-R, which we justify in the following paragraph. With IPA-R, we also use the pretrained encoder parameters $\phi$ as an initialisation for $\psi$, which makes training more stable.

To justify the use of a pretrained $\theta$ and $\phi$ with $\mathcal{O}_{\text{rev}}$, we show that the objective $\mathcal{O}_{\text{rev}}$ has a property analogous to that described by Theorem 3.2.

**Theorem E.2.** *Assume we have a sufficiently expressive encoder and decoder such that there exists parameters $\theta^* \in \Theta$ and $\phi^* \in \Phi$ which make the unconditional VAE objective (Eq. (1)) equal to its upper bound of $-\mathcal{H}\left[p_{\text{data}}(\mathbf{x})\right]$. Then, given a sufficiently expressive partial encoder $c(\mathbf{z}|\mathbf{y}; \psi)$,*

$$\max_{\psi} \mathcal{O}_{rev}(\theta^*, \phi^*, \psi) = \max_{\theta, \phi, \psi} \mathcal{O}_{rev}(\theta, \phi, \psi).$$

**Proof.** We will prove Theorem E.2 by showing that both the quantity on the left-hand side and the quantity on the right-hand side are equal to the negative of the entropy of $p_{\text{data}}(\mathbf{x})$. We begin with the left-hand side,

$$\max_{\psi} \mathcal{O}_{\text{rev}}(\theta^*, \phi^*, \psi) = \max_{\psi} \mathbb{E}_{p_{\text{data}}(\mathbf{y})c(\mathbf{z}|\mathbf{y})} \left[ \log \frac{p^*(\mathbf{z}, \mathbf{y})}{c(\mathbf{z}|\mathbf{y})} \right]. \tag{48}$$

This is a variational lower-bound which is made tight by setting $\psi$ such that $c(\mathbf{z}|\mathbf{y}) = p^*(\mathbf{z}|\mathbf{y})$:

$$\max_{\psi} \mathcal{O}_{\text{rev}}(\theta^*, \phi^*, \psi) = \mathbb{E}_{p_{\text{data}}(\mathbf{y})} \left[ \log p^*(\mathbf{y}) \right] = -\mathcal{H}\left[p_{\text{data}}(\mathbf{y})\right]. \tag{49}$$

Now we consider the right-hand side.

$$\max_{\theta,\phi,\psi} \mathcal{O}_{\mathrm{rev}}(\theta,\phi,\psi) = \max_{\theta,\phi,\psi} \mathbb{E}_{p_{\mathrm{data}}(\mathbf{y})c(\mathbf{z}|\mathbf{y})} \left[ \log \frac{p_{\mathrm{model}}(\mathbf{z},\mathbf{y})}{c(\mathbf{z}|\mathbf{y})} \right] \tag{50}$$

This is another variational lower-bound made tight by setting $\psi$ such that $c(\mathbf{z}|\mathbf{y}) = p_{\mathrm{model}}(\mathbf{z}|\mathbf{y})$:

$$\max_{\theta,\phi,\psi} \mathcal{O}_{\mathrm{rev}}(\theta,\phi,\psi) = \max_{\theta,\phi} \mathbb{E}_{p_{\mathrm{data}}(\mathbf{y})} \left[ \log p_{\mathrm{model}}(\mathbf{y}) \right] \tag{51}$$

$$= -\mathcal{H}\left[ p_{\mathrm{data}}(\mathbf{y}) \right]. \tag{52}$$

The right-hand side is therefore equivalent to the left-hand-side as shown in Eq. (49), proving the theorem.

## F ESTIMATING KL DIVERGENCES IN PRACTICE

In this section we describe how we compute unbiased and low-variance estimates of $\mathcal{O}_{\mathrm{for}}$ and $\mathcal{O}_{\mathrm{rev}}$ in practice. For both, we use similar techniques to those commonly used in unconditional hierarchical VAEs. First note that each can be written as the sum of a likelihood and KL divergence term. For the unconditional VAE objective, $\mathcal{O}_{\mathrm{for}}$ and $\mathcal{O}_{\mathrm{rev}}$ respectively:

$$\mathbb{E}_{p_{\mathrm{data}}(\mathbf{x})} \left[ \mathrm{ELBO}(\theta,\phi,\mathbf{x}) \right] = \mathbb{E}_{p_{\mathrm{data}}(\mathbf{x})} \left[ \mathbb{E}_{q(\mathbf{z}|\mathbf{x})} \left[ \log p_{\mathrm{model}}(\mathbf{x}|\mathbf{z}) \right] - \mathrm{KL}\big( q(\mathbf{z}|\mathbf{x}) \, \| \, p_{\mathrm{model}}(\mathbf{z}) \big) \right], \tag{53}$$

$$\mathcal{O}_{\mathrm{for}}(\theta,\phi,\psi) = \mathbb{E}_{p_{\mathrm{data}}(\mathbf{x},\mathbf{y})} \left[ \mathbb{E}_{q(\mathbf{z}|\mathbf{x})} \left[ \log p_{\mathrm{model}}(\tilde{\mathbf{x}}|\mathbf{z}) \right] - \mathrm{KL}(q(\mathbf{z}|\mathbf{x}) \, \| \, c(\mathbf{z}|\mathbf{y})) \right], \tag{54}$$

$$\mathcal{O}_{\mathrm{rev}}(\theta,\phi,\psi) = \mathbb{E}_{p_{\mathrm{data}}(\mathbf{y})} \mathbb{E}_{c(\mathbf{z}|\mathbf{y})} \left[ \log p_{\mathrm{model}}(\mathbf{y}|\mathbf{z}) - \mathrm{KL}(c(\mathbf{z}|\mathbf{y}) \, \| \, p_{\mathrm{model}}(\mathbf{z})) \right]. \tag{55}$$

In a non-hierarchical VAE with Gaussian $p_{\mathrm{model}}(\mathbf{z})$ and $q(\mathbf{z}|\mathbf{x})$, it is common to compute the KL divergence term analytically in order to reduce the variance of the estimate. In hierarchical VAEs where both $p_{\mathrm{model}}(\mathbf{z})$ and $q(\mathbf{z}|\mathbf{x})$ consists of multiple Gaussian distributions with non-linear dependencies, it is not possible to compute the KL divergence analytically. However, we can still make use of analytic estimates of the KL divergence at each layer. For the unconditional VAE objective, an estimator can be derived as follows:

$$\mathrm{KL}\big( q(\mathbf{z}|\mathbf{x}) \, \| \, p_{\mathrm{model}}(\mathbf{z}) \big) = \mathbb{E}_{q(\mathbf{z}|\mathbf{x})} \left[ \log \frac{q(\mathbf{z}|\mathbf{x})}{p_{\mathrm{model}}(\mathbf{z})} \right] \tag{56}$$

$$= \mathbb{E}_{q(\mathbf{z}|\mathbf{x})} \left[ \log \frac{\prod_{l=1}^{L} q(\mathbf{z}_l|\mathbf{z}_{<l},\mathbf{x})}{\prod_{l=1}^{L} p_{\mathrm{model}}(\mathbf{z}_l|\mathbf{z}_{<l})} \right] \tag{57}$$

$$= \mathbb{E}_{q(\mathbf{z}|\mathbf{x})} \left[ \sum_{l=1}^{L} \log \frac{q(\mathbf{z}_l|\mathbf{z}_{<l},\mathbf{x})}{p_{\mathrm{model}}(\mathbf{z}_l|\mathbf{z}_{<l})} \right] \tag{58}$$

$$= \mathbb{E}_{q(\mathbf{z}|\mathbf{x})} \left[ \sum_{l=1}^{L} \log \frac{q(\mathbf{z}_l|\mathbf{z}_{<l},\mathbf{x})}{p_{\mathrm{model}}(\mathbf{z}_l|\mathbf{z}_{<l})} \right] \tag{59}$$

$$= \sum_{l=1}^{L} \mathbb{E}_{q(\mathbf{z}_{\le l}|\mathbf{x})} \left[ \log \frac{q(\mathbf{z}_l|\mathbf{z}_{<l},\mathbf{x})}{p_{\mathrm{model}}(\mathbf{z}_l|\mathbf{z}_{<l})} \right] \tag{60}$$

$$= \sum_{l=1}^{L} \mathbb{E}_{q(\mathbf{z}_{<l}|\mathbf{x})} \left[ \mathrm{KL}\big( q(\mathbf{z}_l|\mathbf{z}_{<l},\mathbf{x}) \, \| \, p_{\mathrm{model}}(\mathbf{z}_l|\mathbf{z}_{<l}) \big) \right]. \tag{61}$$

This is used by Vahdat & Kautz (2020); Child (2020) to estimate the KL divergence term for an unconditional hierarchical VAE, by sampling $\mathbf{z} \sim q(\cdot|\mathbf{x})$ and then computing the resulting KL divergence for each layer $l$ conditioned on $\mathbf{z}_{<l}$. Similar derivations suggest the following estimates for the KL divergences in $\mathcal{O}_{\mathrm{for}}$ and $\mathcal{O}_{\mathrm{rev}}$:

$$\mathrm{KL}\big( q(\mathbf{z}|\mathbf{x}) \, \| \, c(\mathbf{z}|\mathbf{y}) \big) = \sum_{l=1}^{L} \mathbb{E}_{q(\mathbf{z}_{<l}|\mathbf{x})} \left[ \mathrm{KL}\big( q(\mathbf{z}_l|\mathbf{z}_{<l},\mathbf{x}) \, \| \, c(\mathbf{z}_l|\mathbf{z}_{<l},\mathbf{y}) \big) \right], \tag{62}$$

$$\mathrm{KL}\big( c(\mathbf{z}|\mathbf{y}) \, \| \, p_{\mathrm{model}}(\mathbf{z}) \big) = \sum_{l=1}^{L} \mathbb{E}_{c(\mathbf{z}_{<l}|\mathbf{y})} \left[ \mathrm{KL}\big( c(\mathbf{z}_l|\mathbf{z}_{<l},\mathbf{y}) \, \| \, p_{\mathrm{model}}(\mathbf{z}_l|\mathbf{z}_{<l}) \big) \right]. \tag{63}$$

Table 3: Summary of hyperparameters used for training IPA and our baselines. Reported batch sizes are summed over all GPUs.

| Dataset | Method | Learning rate | Batch size | Iterations | Trainable parameters | GPUs | Training time (GPU-hours) |
|---|---|---|---|---|---|---|---|
| CIFAR-10 | ANP | $5 \times 10^{-5}$ | 16 | 700k | 11.5m | V100 | 33 |
| | CE | $2 \times 10^{-4}$ | 64 | 352k | 34m | 2080 Ti | 14 |
| | RFR | $1 \times 10^{-4}$ | 8 | 675k | 31m | V100 | 450 |
| | PIC | $1 \times 10^{-5}$ | 20 | 1800k | 9m | V100 | 430 |
| | CoModGAN | $2 \times 10^{-3}$ | 32 | 781k | 82m | (2×) V100 | 268 |
| | IPA-R | $2 \times 10^{-5}$ | 16 | 250k | 18m | 2080 Ti | 83 |
| | IPA from scratch | $2 \times 10^{-4}$ | 9 | 3050k | 63m | 2080 Ti | 542 |
| | IPA from ImageNet | $1.5 \times 10^{-4}$ | 16 | 700k | 18m | 2080 Ti | 144 |
| | IPA | $2 \times 10^{-4}$ | 30 | 550k | 18m | V100 | 115 |
| ImageNet-64 | PIC | $1 \times 10^{-5}$ | 20 | 2100k | 9m | V100 | 430 |
| | CoModGAN | $2 \times 10^{-3}$ | 32 | 781k | 99m | (4×) V100 | 552 |
| | IPA-R | $5 \times 10^{-4}$ | 1 | 810k | 58m | 2080 Ti | 121 |
| | IPA from scratch | $5 \times 10^{-5}$ | 1 | 1930k | 183m | 2080 Ti | 226 |
| | IPA | $5 \times 10^{-5}$ | 4 | 700k | 58m | 2080 Ti | 120 |
| FFHQ-256 | ANP | $1 \times 10^{-4}$ | 16 | 990k | 11.5m | V100 | 43 |
| | CE | $2 \times 10^{-4}$ | 64 | 492k | 71.5m | 2080 Ti | 46 |
| | RFR | $1 \times 10^{-4}$ | 8 | 670k | 31m | V100 | 450 |
| | PIC | $1 \times 10^{-5}$ | 20 | 1900k | 9m | V100 | 430 |
| | CoModGAN | $2 \times 10^{-3}$ | 32 | 781k | 108m | (4×) V100 | 1332 |
| | IPA-R | $5 \times 10^{-5}$ | 8 | 416k | 65m | (4×) V100 | 800 |
| | IPA | $1.5 \times 10^{-4}$ | 12 | 626k | 65m | (4×) V100 | 1132 |
| Edges2Bags | IPA from scratch | $5 \times 10^{-5}$ | 1 | 760k | 183m | 2080 Ti | 175 |
| | IPA from ImageNet | $2 \times 10^{-4}$ | 4 | 260k | 58m | 2080 Ti | 45 |
| Edges2Shoes | IPA from scratch | $5 \times 10^{-5}$ | 1 | 750k | 183m | 2080 Ti | 174 |
| | IPA from ImageNet | $2 \times 10^{-4}$ | 4 | 250k | 58m | 2080 Ti | 43 |
| Chest X-ray | CoModGAN | $2 \times 10^{-3}$ | 32 | 265k | 108m | (4×) 2080 Ti | 648 |
| | IPA | $1.5 \times 10^{-4}$ | 8 | 180k | 65m | (4×) V100 | 240 |

We compute unbiased estimates of the KL divergence terms by simply sampling $\mathbf{z}$ (from $q(\cdot|\mathbf{x})$ for $\mathcal{O}_{\text{for}}$ or $c(\cdot|\mathbf{y})$ for $\mathcal{O}_{\text{rev}}$) and analytically computing the resulting KL divergence for each layer $l$ conditioned on $\mathbf{z}_{<l}$.

## G   EXPERIMENTAL DETAILS

### G.1   IPA AND IPA-R INCLUDING FROM-SCRATCH/FROM-IMAGENET VARIANTS

We release code for training IPA and IPA-R, code for using the trained artifacts to perform Bayesian experimental design and out-of-distribution detection, and various pretrained models[1].

**Architectures**   The encoder and decoder architectures we used for CIFAR-10, ImageNet-64, and FFHQ-256 were the same as those used by Child (2020) for the same datasets, with 45, 75, and 62 groups of latent variables respectively. The encoder and decoder had 39 million parameters for CIFAR-10, 125 million for ImageNet-64, and 115 million for FFHQ-256. We used partial encoders with structure identical to the encoders, other than an additional input channel to accept the concatenated mask. The partial encoders contained 18 million, 58 million, and 65 million parameters respectively for CIFAR-10, ImageNet64, and FFHQ-256. The architecture used for the edges-to-photos experiments was the same as that used for ImageNet-64. The architecture used for Chest X-ray 14 was identical to that used for FFHQ-256.

---

[1] https://github.com/plai-group/ipa

**Training details**   Most training hyperparameters were the same as those used by Child (2020) for unconditional training of the corresponding architectures. We report the significant differences in Table 3 and the following paragraph. Learning rates were selected with sweeps over three values, and the batch sizes selected were the largest compatible with the GPU's memory. We use the Weights & Biases (Biewald, 2020) experiment-tracking infrastructure. The only unconditional VAEs which we trained ourselves were for ImageNet-32 (used for the "IPA from ImageNet" runs on CIFAR-10) and the Chest X-ray dataset. We trained the ImageNet-32 VAE on a GeForce RTX 2080 Ti for 14 days, using a batch size of 15 and learning rate of $2 \times 10^{-4}$ (chosen with a sweep over 2 values). We trained the Chest X-ray VAE on 4 V100 GPUs for about 5 days, using a batch size of 8, learning rate of $1.5 \times 10^{-4}$ (chosen with a sweep over 3 values) and "skip threshold[2]" of 15000. While we could have trained a conditional VAE from scratch for the x-ray experiments, training an unconditional model first allowed us to speed up later experimentation with IPA.

### G.2   OTHER BASELINES

For all the remaining baselines, we based our implementations on publicly available (official or unofficial) implementations. A link is provided for each. All the training procedures were modified to use the same distribution of partially observed images as for training IPA (see Section 4).

**CoModGAN**   We used the official implementation of Zhao et al. (2021)[3]. See Table 3 for our hyperparameters, based on those used by Zhao et al. (2021).

**PIC**   We adapted the official implementation of Zheng et al. (2019)[4], and used their reported hyperparameters where possible (see Table 3). The PIC architecture is designed for $256 \times 256$ images. Therefore, in order to test it on ImageNet-64 and CIFAR-10, we resized these images to $256 \times 256$ (via bilinear interpolation) before feeding them into the PIC model. We then down-sampled the inpainted images back to the original size of $32 \times 32$ for evaluation.

**ANP**   Our ANP network architecture was based on that of Kim et al. (2019)[5], differing in that we used hidden and latent dimensions of 512. In the original image inpainting experiments of (Kim et al., 2019) the images are $32 \times 32$. Since ANPs embed each pixel separately and their self-attention and cross-attention layers attend to all other pixels in the observed and target sets, it is computationally expensive to scale them to larger images. We therefore downsampled the FFHQ-256 images to $64 \times 64$, and upsampled the inpainted image back to $256 \times 256$ via bilinear interpolation. Additionally, at training time, we randomly dropped half of the observed pixels (a.k.a. context set in neural process literature) to reduce the computational cost. Finally, the target set at training time was half of the unobserved pixels. For CIFAR-10, on the other hand, no modification was done to the image resolution or observation masks i.e. images were fed in at $32 \times 32$ resolution, the context sets were the set of all observed pixels in masked images $\mathbf{y}$, and the target sets consisted of all unobserved pixels.

**CE**   We use the architecture reported by Pathak et al. (2016)[6].

**RFR**   We adapted the official implementation of Li et al. (2020a)[7]. Similar to PIC, all the input images to this model were resized to $256 \times 256$. The iterative way of completing images in this baseline makes it slow to complete images with most of their pixels missing. Therefore, after around 450

---

[2]Child (2020) improve training stability by skipping updates with gradients larger than a set threshold. Using a larger threshold, and so skipping fewer updates, was found to be necessary to train their unconditional VAE on the Chest X-ray dataset. In all other cases, our "skip threshold" is the same as that used by Child (2020) for the relevant architecture

[3]https://github.com/zsyzzsoft/co-mod-gan

[4]https://github.com/lyndonzheng/Pluralistic-Inpainting

[5]Our implementation of ANP is based on https://github.com/EmilienDupont/neural-processes and https://github.com/YannDubs/Neural-Process-Family

[6]Our implementation of CE is based on https://github.com/BoyuanJiang/context_encoder_pytorch, with some modifications according to https://github.com/pathak22/context-encoder to support larger image sizes and non-centered observation masks.

[7]https://github.com/jingyuanli001/RFR-Inpainting

hours of training on a Tesla V100 GPU, they were trained for only 120 and 85 epochs respectively for CIFAR-10 and FFHQ-256. This may partly explain the poor performance of RFR that we report.

**VQ-VAE**   We adapted the official implementation of Peng et al. (2021)[8]. Similar to PIC and RFR, all the input images for this model were resized to $256 \times 256$. The model has three modules: a VAE, a structure generator and a texture generator. Each is trained separately:

- The VAEs for both datasets were trained for 1 million iterations on an RTX 2080 Ti GPU. It took less than 13 hours for each dataset.
- The structure generator for CIFAR-10 was trained for 1 million iterations and the FFHQ-256 one was trained for 1.5 million iterations. It took 128 hours on a Tesla V100 and 212 hours on an RTX 2080 Ti for CIFAR-10 and FFHQ-256 respectively.
- The texture generators were trained for 2 million traces for both datasets. It took 171 hours on a Tesla V100 and 211 hours on an RTX 2080 Ti GPU, for CIFAR-10 and FFHQ-256 respectively.

This model is very slow to complete images at test time. The authors reported that it takes 45 seconds to complete a $256 \times 256$ image, and in our experiments we observed that this time could be up to 3 minutes on an RTX 2080 Ti GPU. This makes evaluating the quantitative metrics we use prohibitively expensive. We do, however, report qualitative results in Appendix L.

### G.3   METRICS

**FID for completed images**   We use the FID score (Heusel et al., 2017) to quantify the distance between $p_{\text{data}}(\mathbf{x})$ and $\int p_{\text{cond}}(\mathbf{x}|\mathbf{y})p_{\text{data}}(\mathbf{y})\mathrm{d}\mathbf{y}$, the distribution resulting from sampling a dataset image, masking out some pixels, and replacing them by performing inpainting. This should be zero if $p_{\text{cond}}(\mathbf{x}|\mathbf{y}) = p_{\text{data}}(\mathbf{x}|\mathbf{y})$. Since this metric does not explicitly consider multiple completions of the same observation, we view it as a measure of image quality more than diversity.

**P-IDS**   This metric was proposed by Zhao et al. (2021), who show that is correlates well with human evaluations. It measures the linear separability of dataset and inpainted images in a pretrained Inception network's feature space. Higher values are better, indicating that the data is less separable.

**LPIPS diversity score**   This metric for diversity was proposed by Zhu et al. (2017), involving calculating the LPIPS distance between pairs of images sampled from $p_{\text{cond}}(\cdot|\mathbf{y})$ for the same $\mathbf{y}$. We report it here for completeness but it is not a good measure of the "fit" between $p_{\text{cond}}(\mathbf{x}|\mathbf{y})$ and $p_{\text{data}}(\mathbf{x}|\mathbf{y})$ since it can always be maximised by making $p_{\text{cond}}(\mathbf{x}|\mathbf{y})$ as diverse as possible, without reference to the "true" posterior $p_{\text{data}}(\mathbf{x}|\mathbf{y})$.

**Faithfulness weighted variance**   The faithfulness weighted variance (FWV) was proposed by Li et al. (2020b), measuring both how diverse generated images $\mathbf{x} \sim p_{\text{cond}}(\cdot|\mathbf{y})$ are and how well they fit the ground truth. To compute it we first draw $N$ pairs of $(\mathbf{x}_i, \mathbf{y}_i)$ from the test set. Then for each $\mathbf{y}_i$ we draw $K$ conditional samples from the model $\hat{\mathbf{x}}_{i,j} \sim p_{\text{cond}}(\cdot|\mathbf{y}_i)$ and let $\bar{\mathbf{x}}_i$ be the pixel-wise average of $\{\hat{\mathbf{x}}_{i,j}\}_{j=1}^{K}$. Finally,

$$\text{FWV} = \sum_{i=1}^{N}\sum_{j=1}^{K} w_{i,j} d_{\text{LPIPS}}(\hat{\mathbf{x}}_{i,j}, \bar{\mathbf{x}}_i), \text{ where } w_{i,j} = \exp\left(-\frac{d_{\text{LPIPS}}(\hat{\mathbf{x}}_{i,j}, \mathbf{x}_i)}{2\sigma^2}\right). \quad (64)$$

We normalise by $N$ and report $\frac{\text{FWV}}{N}$. We use $N = K = 100$ and compute the FWV for various values of $\sigma$. The parameter $\sigma$ determines how closely a sample must match the ground truth to be able to contribute to the FWV. Using a low $\sigma$ therefore favours methods which can most faithfully reconstruct the ground truth, and using a high $\sigma$ favours methods which produce diverse samples, regardless of their match to the ground truth. We compute this metric for inpainting tasks, in which generated images are typically much more diverse than in the super-resolution task it was designed for (Li et al., 2020b). We therefore report scores orders of magnitude higher than Li et al. (2020b) and note that it is unclear how meaningful the pixel-space average is in our setting.

---

[8]https://github.com/USTC-JialunPeng/Diverse-Structure-Inpainting

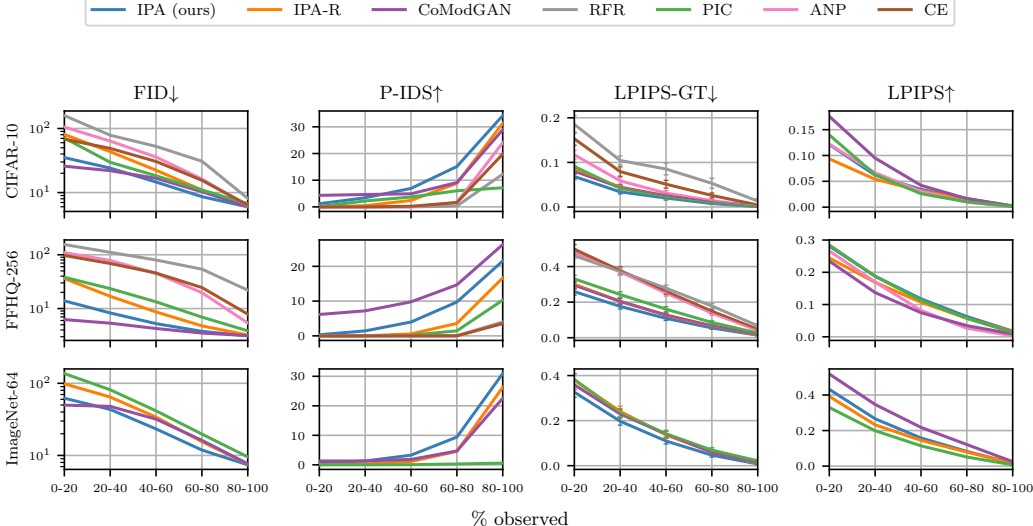

Figure 6: Expanded Fig. 3 including results on ImageNet-64 and the LPIPS diversity score.

## H ADDITIONAL RESULTS

Figure 6 shows a breakdown of various metric for different sizes of mask. This is similar to Fig. 3 in the main paper, but also shows the LPIPS diversity score and scores on ImageNet-64. In Fig. 7, we plot the faithfulness weighted variance described in Appendix G.3. We compute it for the values of $\sigma$ reported by Li et al. (2020b), as well as two higher values and two lower values.

## I BAYESIAN OPTIMAL EXPERIMENTAL DESIGN

### I.1 EIG ESTIMATORS

In this section, we expand on how our estimator for the expected information gain differs from prior work (Harvey et al., 2019). Repeating some relevant definitions from the main text, $f_{l_1,\dots,l_t}$ is a function which maps from a full image to a sequence of observations of the image at locations $l_1, \dots, l_t$. When estimating the EIG at time step $t$, we will have already observed a sequence of observations denoted $\mathbf{y}_{l_1,\dots,l_{t-1}}$, extracted from some latent image $\mathbf{x}$ as $f_{l_1,\dots,l_{t-1}}(\mathbf{x})$. We use a CNN which outputs $g(v|f_{l_1,\dots,l_t}(\mathbf{x}))$, an approximation of the posterior over the latent variable of interest given a sequence of observations. Following Harvey et al. (2019), we will from now on refer to this as the AVP-CNN (attentional variational posterior CNN). Both our method and that of Harvey et al. (2019) sample image completions, which we denote $\mathbf{x}^{(1)}, \dots, \mathbf{x}^{(N)}$, conditioned on $\mathbf{y}_{l_1,\dots,l_{t-1}}$. Harvey et al. (2019) do so by retrieving images which roughly match the observed pixel values from a database. Although completions from this are diverse, they can match the observed values poorly. We therefore replace this stage with an image completion network trained using IPA.

Given these components, Harvey et al. (2019) approximate the expected information gain with

$$\text{EIG}(l_t; \mathbf{y}_{l_1,\dots,l_{t-1}}) \approx \overbrace{\mathcal{H}\left[g(\cdot|\mathbf{y}_{l_1,\dots,l_{t-1}})\right]}^{\text{entropy after } t-1 \text{ scans}} - \frac{1}{N}\sum_{n=1}^{N} \overbrace{\mathcal{H}\left[g(\cdot|f_{l_1,\dots,l_t}(\mathbf{x}^{(n)}))\right]}^{\text{expected entropy after } t \text{ scans}}. \quad (65)$$

Since the prior entropy term of Eq. (65) does not depend on $l_t$, it can be neglected when choosing $l_t$ with BOED. As only the expected posterior entropy then needs to be estimated and compared for various $l_t$, we will from now on refer to this estimator with the acronym EPE (for 'expected posterior entropy').

We use a different estimator for the prior entropy. It becomes equivalent as $N \to \infty$ if $g$ and the image completion method are perfect but, when this is not the case, produces an estimate of the prior

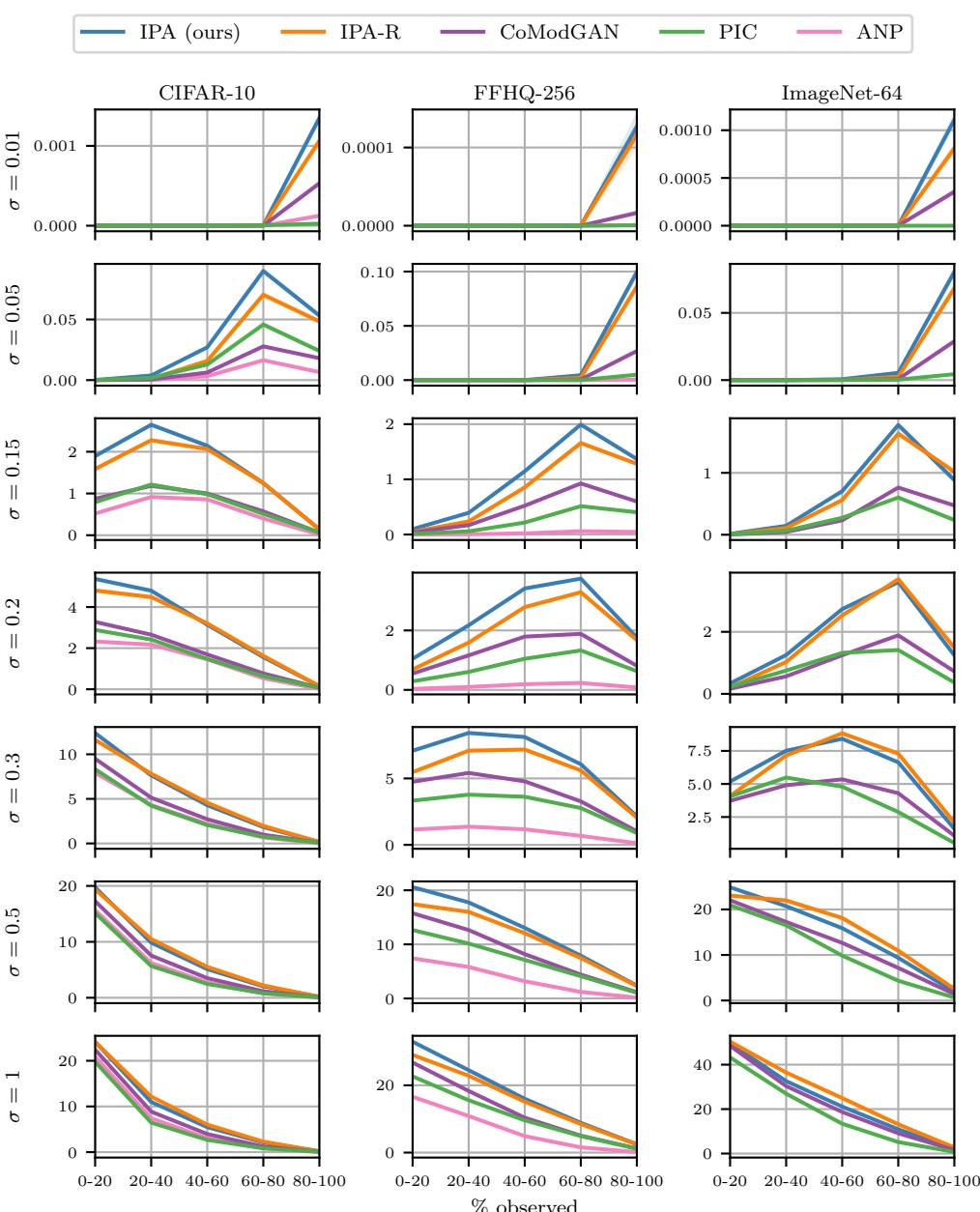

Figure 7: The faithfulness weighted variance for inpainting on CIFAR-10, FFHQ-256, and ImageNet-64, for various values of $\sigma$. IPA obtains the best performance on almost all datasets and values of $\sigma$, only being outperformed by IPA-R on CIFAR-10 with high values of $\sigma$. This indicates that IPA both assigns high probability density to the ground truth (and so performs well for small $\sigma$) and generates diverse samples (and so performs well for larger $\sigma$).

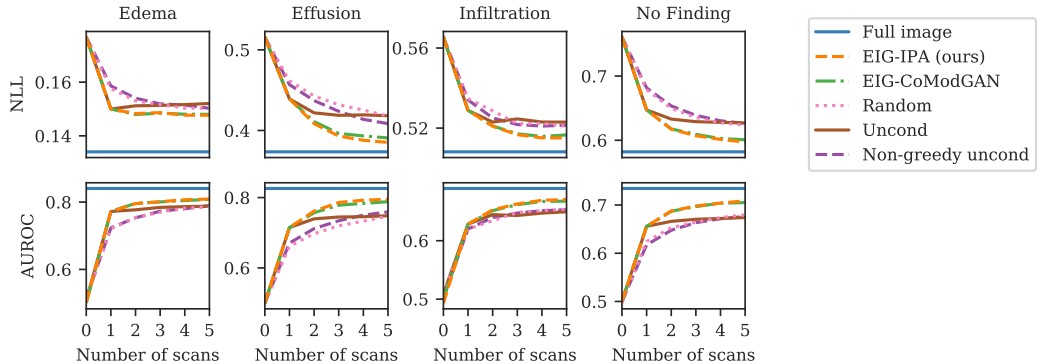

Figure 8: We extend on Fig. 5 by additionally reporting negative log-likelihoods (top row) for the classification tasks. We also include results from the 'EPE' estimator used by Harvey et al. (2019). We see that our BOED estimator provides significant improvements; resulting performance is always superior to the other baselines. This is only sometimes the case for the 'EPE' estimator.

entropy with a dependence on $l_t$. To repeat Eq. (10), this results in the following estimator for the EIG:

$$\mathrm{EIG}(l_t; \mathbf{y}_{l_1, \ldots, l_{t-1}}) \approx \overbrace{\mathcal{H}\left[\frac{1}{N}\sum_{n=1}^{N} g(\cdot | f_{l_1, \ldots, l_t}(\mathbf{x}^{(n)}))\right]}^{\text{entropy after } t-1 \text{ scans}} - \overbrace{\frac{1}{N}\sum_{n=1}^{N}\mathcal{H}\left[g(\cdot | f_{l_1, \ldots, l_t}(\mathbf{x}^{(n)}))\right]}^{\text{expected entropy after } t \text{ scans}}. \quad (66)$$

While it is not immediately clear that this estimator is better than that in Eq. (65), one useful property is that, like the true expected information gain, it is guaranteed to be non-negative. This can be seen as follows. First, note that the prior entropy term is the entropy of an expectation (over $n \sim \mathrm{Uniform}(1, N)$), and the expected posterior entropy term swaps the order of the entropy and expectation. Since the mapping from a distribution $g(\cdot)$ to its entropy $\mathcal{H}[g(\cdot)]$ is strictly concave, Jensen's inequality is applicable. Invoking Jensen's inequality shows that the prior entropy term must be greater than or equal to the expected posterior entropy, and so the estimate of the EIG will be non-negative. In Fig. 8, we compare the performance of this estimator (denoted EIG) and that in Eq. (65) (EPE). We find that the estimator denoted EIG leads to significantly better classification performance.

Another view of our EIG estimator is as a variant of the inception score (Salimans et al., 2016) for the sampled images $\mathbf{x}^{(1)}, \ldots, \mathbf{x}^{(N)}$. The standard inception score is computed using an image classifier trained on ImageNet (Salimans et al., 2016). Equation (10) is the inception score if this classifier is replaced with the AVP-CNN acting on observations of each image at locations $l_1, \ldots, l_t$.

## I.2 BOED BASELINES

**Random** This baseline simply samples the scan location independently at each time step from a uniform distribution over all valid locations.

**Uncond** This baseline ablates our estimator for the EIG (Eq. (10)) by sampling images $\mathbf{x}^{(1)}, \ldots, \mathbf{x}^{(N)}$ from the training dataset (without conditioning on $\mathbf{y}_{l_1, \ldots, l_{t-1}}$) instead of from IPA (conditioned on $\mathbf{y}_{l_1, \ldots, l_{t-1}}$). That is, we use:

$$\mathrm{U}^{\mathrm{uncond}}(l_t; l_1, \ldots, l_{t-1}) = \mathcal{H}\left[\frac{1}{N}\sum_{n=1}^{N} g(\cdot | f_{l_1, \ldots, l_t}(\mathbf{x}^{(n)}))\right] - \frac{1}{N}\sum_{n=1}^{N}\mathcal{H}\left[g(\cdot | f_{l_1, \ldots, l_t}(\mathbf{x}^{(n)}))\right] \quad (67)$$

with $\mathbf{x}^{(1)}, \ldots, \mathbf{x}^{(N)} \sim p(\mathbf{x})$. The resulting choice of scan location $l_t$ at each time step $t$ has no dependence on previous observations $\mathbf{y}_{l_1, \ldots, l_{t-1}}$.

with locations chosen to infer

| | | Edema | Effusion | Infiltration | No Finding |
|---|---|---|---|---|---|
| AUROC when predicting | Edema | 0.807 | 0.801 | 0.804 | 0.804 |
| | Effusion | 0.783 | 0.791 | 0.781 | 0.781 |
| | Infiltration | 0.674 | 0.671 | 0.675 | 0.675 |
| | No Finding | 0.704 | 0.707 | 0.708 | 0.708 |

Figure 9: AUROC scores when performing one classification task with scan locations chosen for a different classification task. The intensity of the colour of each cell is proportional to the difference between its value and the greatest value in its row.

**Non-greedy uncond** When selecting scan locations that optimise the EIG, we select the scan location at each time step greedily to optimise the EIG from the next scan. To do otherwise (i.e. maximise the EIG from multiple next scans) is intractable because of the dependence of the EIG at each $t$ on previous observations $\mathbf{y}_{l_1,\dots,l_{t-1}}$. However, since the above 'Uncond' baseline selects each $l_t$ independently of what is observed at previous scans, we can remove this greedy property. We do so by selecting all $l_1, \dots, l_T$ simultaneously to maximise

$$\mathrm{U}^{\text{non-greedy}}(l_1, \dots, l_T) = \mathcal{H}\left[\frac{1}{N}\sum_{n=1}^{N} g(\cdot | f_\emptyset(\mathbf{x}^{(n)}))\right] - \frac{1}{N}\sum_{n=1}^{N}\mathcal{H}\left[g(\cdot | f_{l_1,\dots,l_T}(\mathbf{x}^{(n)}))\right] \quad (68)$$

with $\mathbf{x}^{(1)}, \dots, \mathbf{x}^{(N)} \sim p(\mathbf{x})$. Since the number of possible values of $l_1, \dots, l_T$ increases exponentially with $T$, it is not feasible to search over all of them. We therefore optimise this sequence by randomly sampling a set of such sequences, and choosing the best from this set.

## I.3 ADDITIONAL BOED RESULTS

Figure 8 shows additional results from classification with sequences chosen by BOED and our baselines. In particular, we report negative log-likelihoods for the class labels as well as the AUROC scores. We see that the negative log-likelihood sometimes increases as more scans are taken, indicating that the classifier used is not perfectly trained. Interestingly, this only happens when glimpses are chosen with the EPE estimator of Harvey et al. (2019). This may be because the EPE estimator has a tendency to select locations which cause the classifier to be overconfident (and thus produce artificially low posterior entropies). With the EIG estimator, and the baselines, this issue is not observed.

To illustrate that the scan locations chosen by BOED are task-dependent, Fig. 9 shows the results from using scan locations chosen for one classification task (i.e. diagnosing a particular disease) to perform a different classification task. Each row shows the results for a particular task, with each column showing results when scan locations are chosen based on a particular task. As expected, the highest (or at least joint-highest) AUROC score in each row occurs on the diagonal, when the classification task and choice of scan locations are aligned. Away from the diagonal, the scores are only slightly lower, perhaps reflecting that there is a large overlap between the relevant areas for these tasks.

Figure 10 shows additional visualisations of the Bayesian experimental design process for different test images, adding to the one shown in the main paper. The 'information gains' shown for a particular $\mathbf{x}^*$ (sampled conditioned on observations $\mathbf{y}_{l_1,\dots,l_{t-1}}$) are computed as the following KL

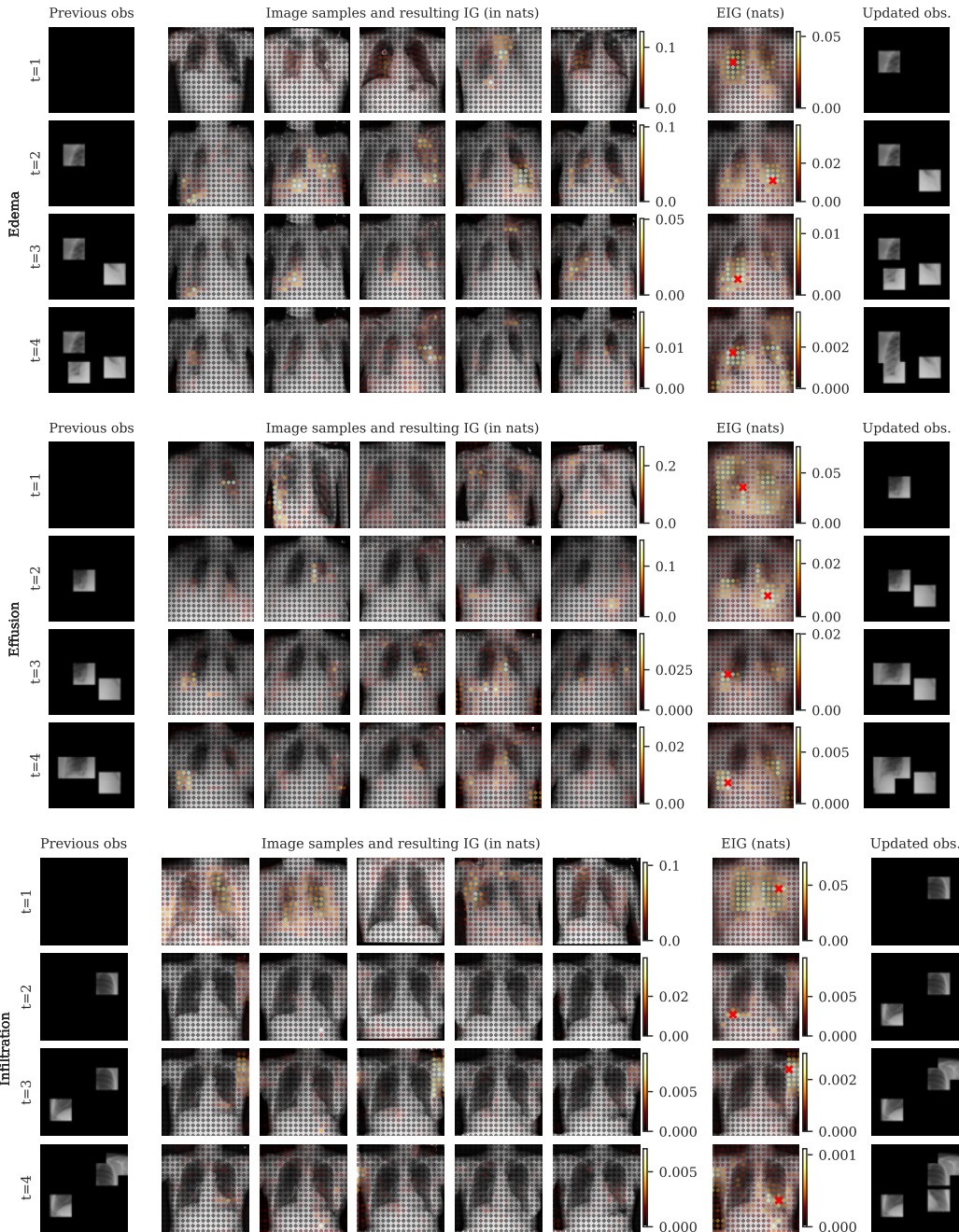

Figure 10: Visualisations of BOED processes. On the top, the task is to detect 'Edema'. In the middle, we aim to detect 'Effusion' in a different patient. The task at the bottom is to detect 'Infiltration'. Four scan locations are selected for each. The left column shows the observations made before selecting each scan location. The next five columns show five of the $N = 10$ sampled image completions conditioned on these observations. Each is overlaid with the information gain predicted after scanning any location. The second column from the left shows the expected information gain at each location, given by averaging the information gains arising from each sampled image completion. A red cross marks the maximum. The final column shows the updated observations after scanning the location which maximises the expected information gain.

divergence:

$$\text{IG}(\mathbf{x}^*, l_t; \mathbf{y}_{l_1,\ldots,l_{t-1}}) \approx \text{KL}\left(g(\cdot|f_{l_1,\ldots,l_t}(\mathbf{x}^*)) \,\middle\|\, \frac{1}{N}\sum_{n=1}^{N} g(\cdot|f_{l_1,\ldots,l_t}(\mathbf{x}^{(n)}))\right). \qquad (69)$$

Crucially, averaging information gains computed with $\mathbf{x}^* = \mathbf{x}^{(1)}$ to $\mathbf{x}^{(N)}$ gives the estimate of the EIG presented in Eq. (10).

### I.4 BOED EXPERIMENTAL DETAILS

The AVP-CNN, $g$, is trained to map from masked images, $\mathbf{y}$, to distributions over the class labels. Each image in the Chest X-ray 14 dataset has labels indicating the presence or absence of each of 14 pathologies. We train $g$ to produce 15 outputs: an estimated probability of the presence or absence of each of the 14 conditions individually; and an additional estimate of the probability that any (one or more) of these conditions is present. We train $g$ to estimate these using a cross-entropy loss. Masked images are sampled using almost the same mask distribution as for training the image completion networks (described in Section 4); the only difference is that patches now have 25% rather than 35% of the image width, to match the observations we use in the experiments with BOED. We use an AVP-CNN pretrained on ImageNet and then trained on Chest X-ray 14 for 32 000 iterations with a batch size of 32 and learning rate $1 \times 10^{-5}$. We train IPA as described in Appendix G.

We select each scan location by evaluating Eq. (10) at every point in an evenly-spaced $17 \times 17$ grid over the image, and choosing the maximum. We evaluate the EIG with $N = 10$ sampled image completions. This is repeated to select the scan location for each $t = 1, \ldots, 5$ (with the sampled images conditioned on observations up to $t - 1$ at each stage). Where applicable, we use the same trained networks and hyperparameters for the baselines. For the 'non-greedy uncond' baseline, we select a scan location sequence by choosing the best of 289 sampled sequences. This number was chosen to match the number of locations in the $17 \times 17$ grid that the other methods search over at each time step. The results reported in Figs. 5 and 8 were computed on a randomly-sampled 5000 image subset of the Chest X-ray 14 test set.

## J OUT-OF-DISTRIBUTION DETECTION

A major advantage of likelihood-based models is the possibility to use them to detect when an input is dissimilar to the training data (out-of-distribution, or OOD). Since a learned model is likely to perform poorly on such inputs, it is important for deployed systems to be able to recognise them and this has been the focus of a substantial body of work (Ren et al., 2019; Hendrycks & Gimpel, 2016; Xiao et al., 2020; Havtorn et al., 2021). The task we consider is detecting whether a partially observed image (e.g. the input to an image completion system) is OOD.

OOD-detection using probabilistic models is theoretically appealing and robust, since no assumptions need to be made about the OOD data (Xiao et al., 2020; Havtorn et al., 2021). Such OOD-detection metrics commonly have the following interface: they take as input a probabilistic model (e.g. a VAE) and a data point, and output a scalar value. The greater the value of this scalar, the more likely it is that the data point is within the distribution of the generative model. One intuitive metric is simply the log-likelihood of the data point under the model. This can work poorly in practice, however, and it has been found that learned models sometimes assign higher likelihood to out-of-distribution data than to in-distribution data (Nalisnick et al., 2018). Several alternative OOD-detection metrics have been proposed specifically for VAEs (Xiao et al., 2020; Havtorn et al., 2021). To use them, we note that we can use networks trained with either IPA or IPA-R to lower-bound the likelihood of a partially-observed image. Specifically, this lower-bound is the reverse KL objective in Eq. (42). After conducting preliminary experiments using the VD-VAE with several OOD-detection metrics (Xiao et al., 2020; Havtorn et al., 2021), we found that improved performance could be obtained (for OOD-detection of both partially- and fully-observed images) using the "temperature gradient" metric described further in Appendix J.1 below. It is efficient to compute, requiring the VAE to be run just once and gradients computed.

As a baseline, we complete the partially observed image with our best performing baseline image completion method, CoModGAN, and perform OOD-detection on the result (using the "temperature gradient" metric on an unconditional VD-VAE). Figure 11 shows our results on CIFAR-10 and

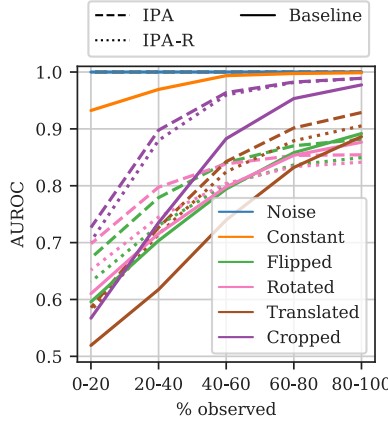

| ID | CIFAR-10 | | | | |
|---|---|---|---|---|---|
| OOD | Noise | Const. | Flip | Rotate | SVHN |
| IPA | **.995** | **.949** | .552 | .594 | **.753** |
| IPA-R | .987 | .921 | **.635** | **.619** | .686 |
| Baseline | .993 | .909 | .543 | .536 | .556 |

| ID | FFHQ-256 | | | | |
|---|---|---|---|---|---|
| OOD | Noise | Const. | Flip | Rotate | Crop |
| IPA | **1.00** | **1.00** | **.809** | **.808** | **.912** |
| IPA-R | **1.00** | **1.00** | .770 | .775 | .902 |
| Baseline | **1.00** | 0.98 | .769 | .771 | .823 |

Figure 11: OOD-detection on incomplete images. On the left, we show AUROC scores on FFHQ-256 computed separately for masks from varying distributions. On the right we report, for both CIFAR-10 and FFHQ-256, the average of these scores over all mask distributions.

FFHQ-256. We test various types of OOD data: **Noise** samples each pixel value independently from a uniform distribution; **Constant** uniformly samples a single colour for the entire image; **Flip** vertically flips an in-distribution (ID) image; **Rotate** rotates an ID image by 90°; **Crop** upsamples an ID image by a factor of 1.5 and then crops to the original size. **SVHN** is the Street View House Number dataset (Netzer et al., 2011). With 80-100% of the image observed, we obtain better AUROCs than similar methods applied to the full image. In particular, on CIFAR-10 vs SVHN, we outperform both Xiao et al. (2020) and Havtorn et al. (2021), albeit with a larger VAE architecture. Using IPA for OOD-detection results in performance that degrades gracefully as less of the image is observed. There are therefore particularly large benefits over the baseline when only 0-20% or 20-40% of the image is observed.

Figure 12 shows a breakdown of the AUROC scores for each mask distribution on CIFAR-10, similar to that shown for FFHQ-256 in Fig. 11. Error bars are standard deviations computed with 3 runs, with the unconditional VD-VAE and IPA/IPA-R retrained each time.

## J.1 OUT-OF-DISTRIBUTION DETECTION WITH TEMPERATURE GRADIENTS

We conducted preliminary experiments with several techniques for OOD-detection with VAEs, applying each to the VD-VAE architecture. Using the likelihood-regret technique (Xiao et al., 2020) with the VD-VAE is computationally costly (as it involves optimising neural network parameters), and we obtained results on CIFAR-10 considerably worse than those produced by the smaller VAE architecture with which it was introduced. We also implemented the log-likelihood ratio metric (Havtorn et al., 2021) for the VD-VAE, but did not find any hyperparameter configurations with which it was consistently better than random guessing across different tasks.

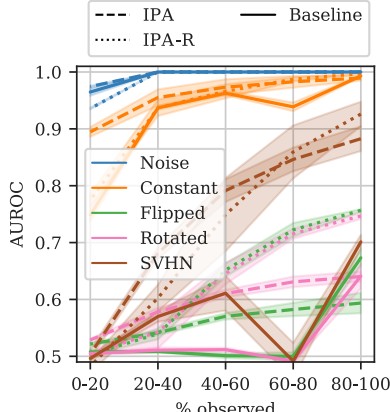

Figure 12: Breakdown of AUROC for each mask distribution on CIFAR-10.

We therefore introduce a metric which we found to work well with the VD-VAE. It is based on a common heuristic to make a VAE produce more regular samples: reducing the variance of the prior over its latent variables (Vahdat & Kautz, 2020). If $\mathbf{t}$ is a vector containing the temperature $\mathbf{t}_l$ for each layer $l$, then doing so corresponds to modifying the prior $p_{\text{model}}(\mathbf{z})$ so that the Gaussian

distribution over each group of latent variables is scaled by the corresponding temperature:

$$p_{\text{model}}(\mathbf{z}; \mathbf{t}) = \prod_{l=1}^{L} \frac{1}{C(\mathbf{t}_l, \mathbf{z}_{<l})} p_{\text{model}}(\mathbf{z}_l | \mathbf{z}_{<l})^{\frac{1}{\mathbf{t}_l^2}} \tag{70}$$

where $C(\mathbf{t}_l, \mathbf{z}_{<l})$ is a normalisation constant. In practice, since $p_{\text{model}}(\mathbf{z}_l | \mathbf{z}_{<l})$ is a Gaussian distribution, this scaling corresponds to simply multiplying the standard deviation by $\mathbf{t}_l$.

We can lower-bound the corresponding marginal likelihood $p_{\text{model}}(\mathbf{y}; \mathbf{t})$ using the partial encoder similarly to the training objective in Eq. (42):

$$\mathbb{E}_{p_{\text{data}}(\mathbf{y})} \left[ \log p_{\text{model}}(\mathbf{y}; \mathbf{t}) \right] \geq \mathbb{E}_{p_{\text{data}}(\mathbf{y})} \mathbb{E}_{c(\mathbf{z}|\mathbf{y})} \left[ \log \frac{p_{\text{model}}(\mathbf{z}; \mathbf{t}) p_{\text{model}}(\mathbf{y}|\mathbf{z})}{c(\mathbf{z}|\mathbf{y})} \right]. \tag{71}$$

We hypothesise that OOD examples will usually either be excessively regular, and therefore have a higher marginal likelihood for $t < 1$, or excessively irregular, and therefore have a higher likelihood for $t > 1$. In-distribution (ID) examples, on the other hand, should have the highest marginal likelihood for $t \approx 1$. A useful feature to distinguish between ID and OOD examples is therefore likely to be the gradient $\frac{\partial}{\partial \mathbf{t}} p(\mathbf{y}; \mathbf{t})|_{\mathbf{t}=\mathbf{1}}$, which we can approximate with the gradient w.r.t. $\mathbf{t}$ of the lower-bound above. We fit a multivariate Gaussian to 5000 samples of $\frac{\partial}{\partial \mathbf{t}} p(\mathbf{y}^{(i)}; \mathbf{t})|_{\mathbf{t}=\mathbf{1}}$ with $\mathbf{y}^{(i)}$ sampled from the training data with the training mask distribution. To obtain a score for how in-distribution a new example $\mathbf{y}$ is, we compute the probability density of $\frac{\partial}{\partial \mathbf{t}} p(\mathbf{y}; \mathbf{t})|_{\mathbf{t}=\mathbf{1}}$ under this multivariate Gaussian. The greater this score, the more likely it is that an example is in-distribution. We call this the "temperature-gradient" metric.

### J.2  OOD DETECTION DETAILS

We compute the AUROC scores presented in Fig. 11 using the 5000 image test set for CIFAR-10 and its OOD-transformations; a 5000 images subset of the SVHN test set; and the 7000 image test set for FFHQ-256. We use 5000 samples of $32 \times 32$ 'Noise' and 'Constant' images when comparing against CIFAR-10, and 200 samples at $256 \times 256$ resolution when comparing against FFHQ-256.

We also present results for the 'Translated' transformation in Fig. 11 for FFHQ-256. This transformation takes a crop with 90% of the image size from one of the four corners and then rescales it to the original size. The result is thus slightly translated so that e.g. the eyes of an FFHQ image are no longer aligned. It is not shown in our results table due to space constraints.

### K  LACK OF SEMANTIC DIVERSITY IN COMODGAN

In this section we demonstrate the some failure cases of CoModGAN in which it fails to produce semantically diverse images. We suspect that this behaviour of CoModGAN contributes to its poor performance on the LPIPS-GT metric. Figure 13 shows a few examples of partial images for which CoModGAN struggles to generate semantically diverse images. Compare these results with IPA's results reported in Fig. 14, showing a wide range of completions for all the partial images.

### L  IMAGE SAMPLES

Figures 15 to 20 on the following pages show conditionally generated images for all datasets we have experimented with, from both IPA and our baselines. It is apparent from the image samples, particularly when the number of observed pixels is small, that IPA-R has less sample diversity than IPA, which can be explained by the mode-seeking behaviour of the reverse-KL and subsequent under-estimation of posterior uncertainty (Minka et al., 2005). Finally, we note a flaw with some of the completions produced by IPA: a lack of bilateral symmetry in FFHQ-256 samples. Again, this is most apparent when few pixels are observed. This issue can be seen in unconditional samples from the underlying VAEs as well (see the samples reported by Child (2020)). Therefore, it is likely that any future advances in image modelling with VAEs could be integrated to improve this aspect of the results.

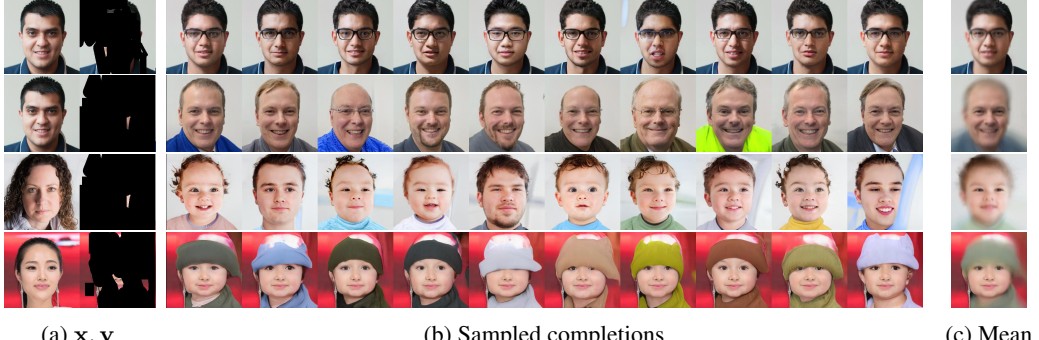

(a) **x**, **y**            (b) Sampled completions            (c) Mean

Figure 13: Examples of completions lacking semantic diversity in CoModGAN. Panel (a) shows the true image and the masked version on which the completions are conditioned. Panel (b) shows 10 completions sampled randomly from the CoModGAN model. Panel (c) shows the mean image computed from 100 sampled completions. On each row, the completions are mostly semantically similar to eachother yet different from the ground truth image, indicating that they are not faithfully representing the true posterior. This behaviour can be contrasted with that of IPA in Fig. 14.

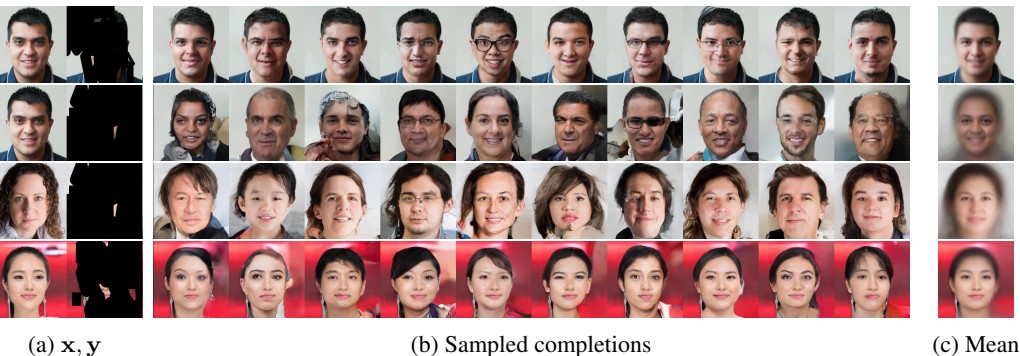

(a) **x**, **y**            (b) Sampled completions            (c) Mean

Figure 14: Examples completions from our IPA model, to compare with Fig. 13. These samples from IPA provide a much better representation of the inherent uncertainty in the image given the observations. We did not find any (**x**, **y**) pairs on which the IPA model failed in the same way as CoModGAN in Fig. 13.

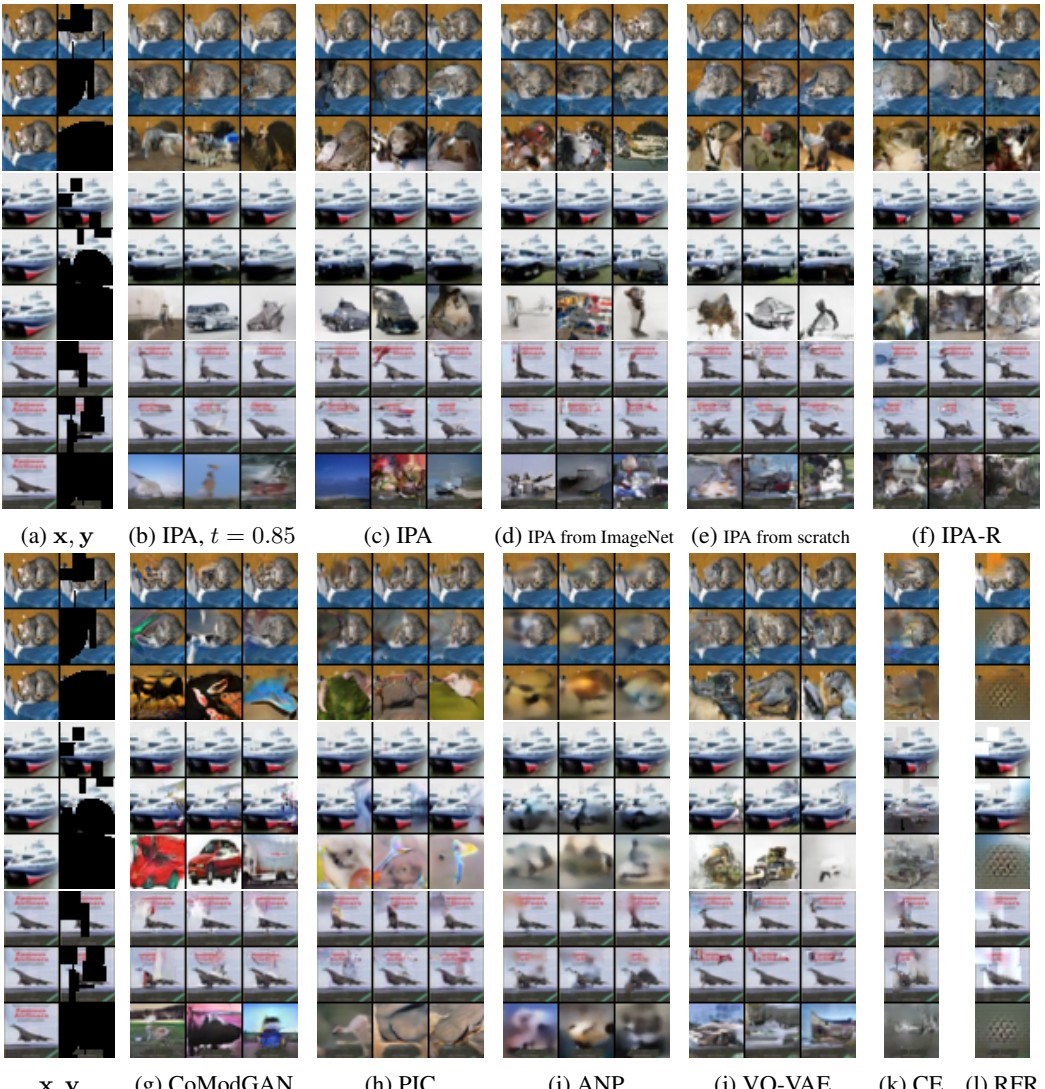

Figure 15: Sampled completions on CIFAR-10. Panel (a) shows test images along with masked versions on which samples in each row are conditioned. The remaining panels in the top row show samples from IPA (with an artifact pretrained on the same dataset) with and without a reduced temperature, IPA with an artifact pretrained on ImageNet, a conditional VAE trained from scratch, and finally IPA-R (with an artifact pretrained on the same dataset). The bottom row shows samples from our baselines. Three samples per masked image are shown for each stochastic method, while the single deterministic completion is shown for CE and RFR. All samples are taken with temperature 1 where not stated otherwise.

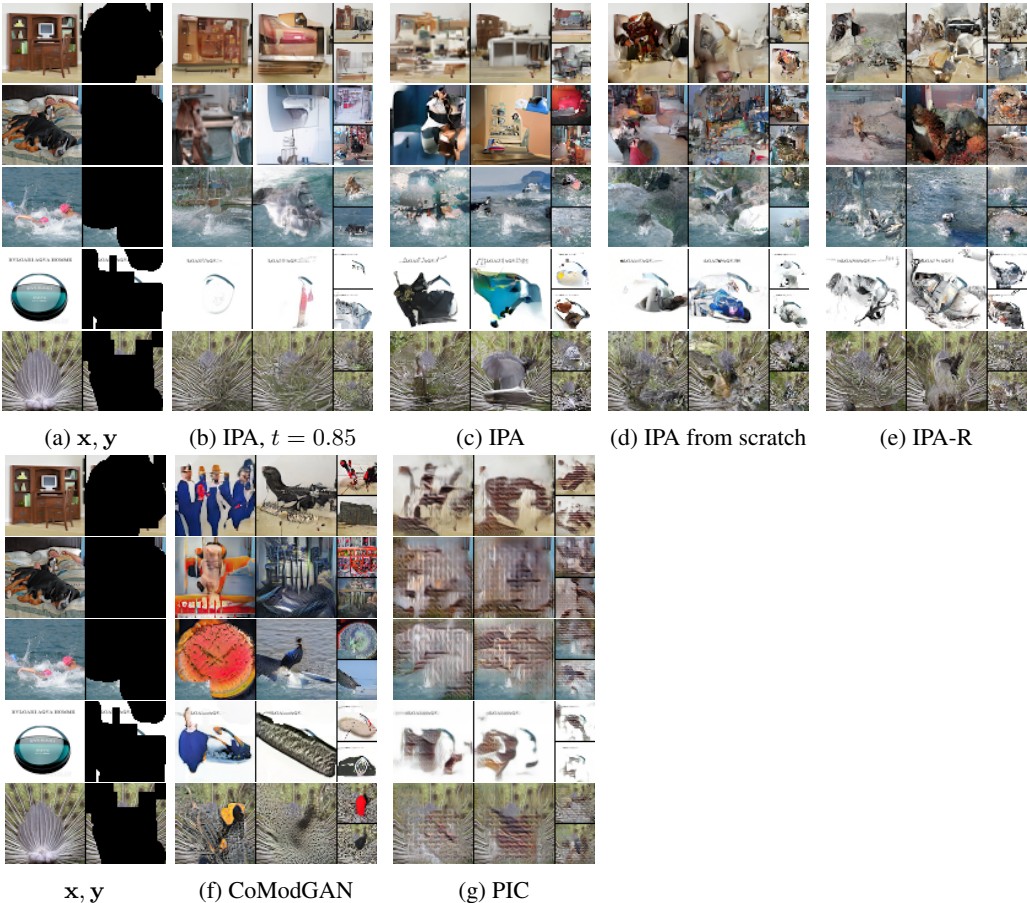

Figure 16: Sampled completions on ImageNet-64. Panel (a) shows a test image and the masked version on which samples in each row are conditioned. The remaining panels in the top row show samples from IPA with and without a reduced temperature, from an IPA-style conditional VAE trained from scratch, and from IPA-R. The bottom row shows samples CoModGAN.

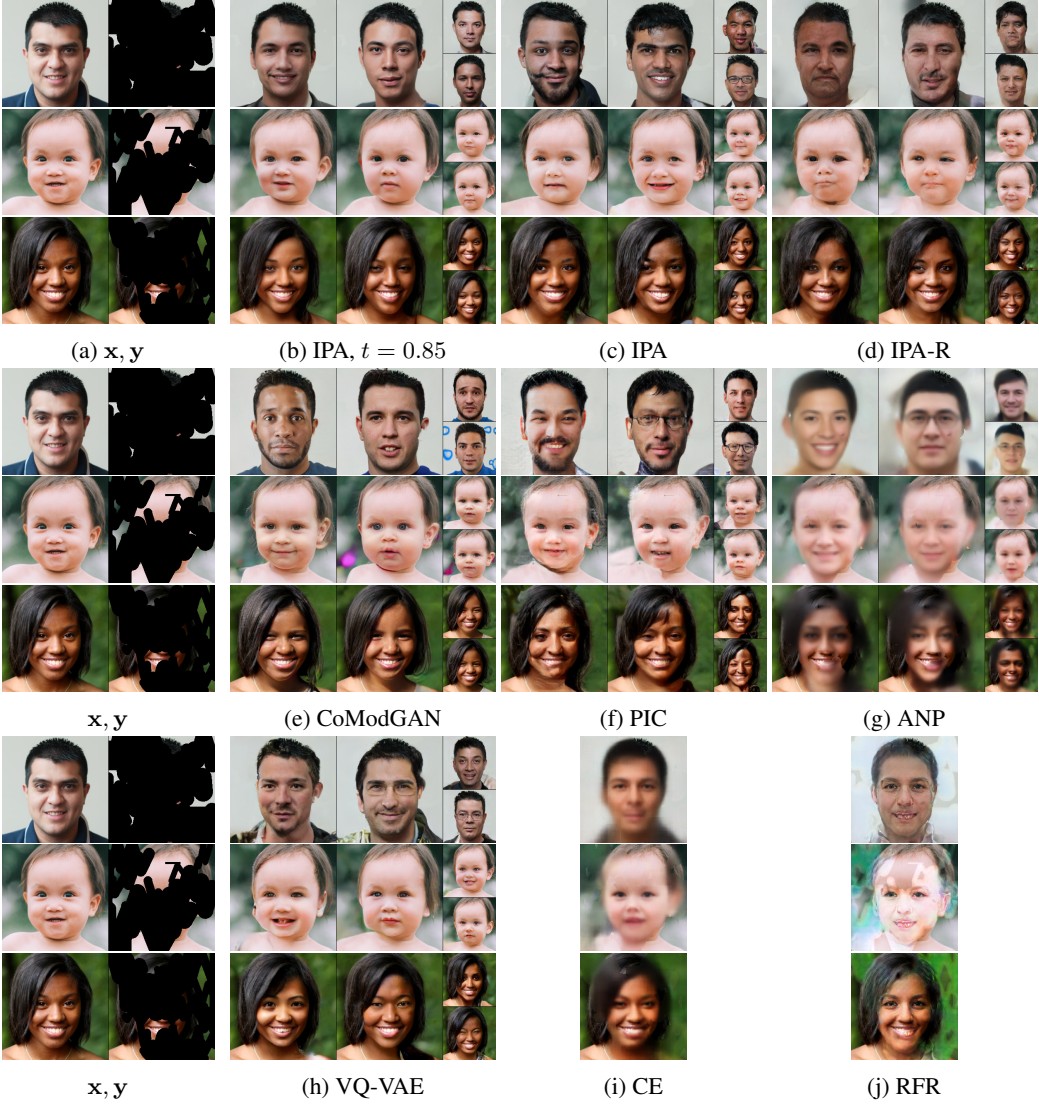

Figure 17: Sampled completions on FFHQ-256. Panel (a) shows test images along with masked versions on which samples in each row are conditioned. The remaining panels in the top row show samples from IPA with and without a reduced temperature and from IPA-R. The other rows show samples from our baselines. Four samples per masked image are shown for each stochastic method, while the single deterministic completion is shown for CE and RFR.

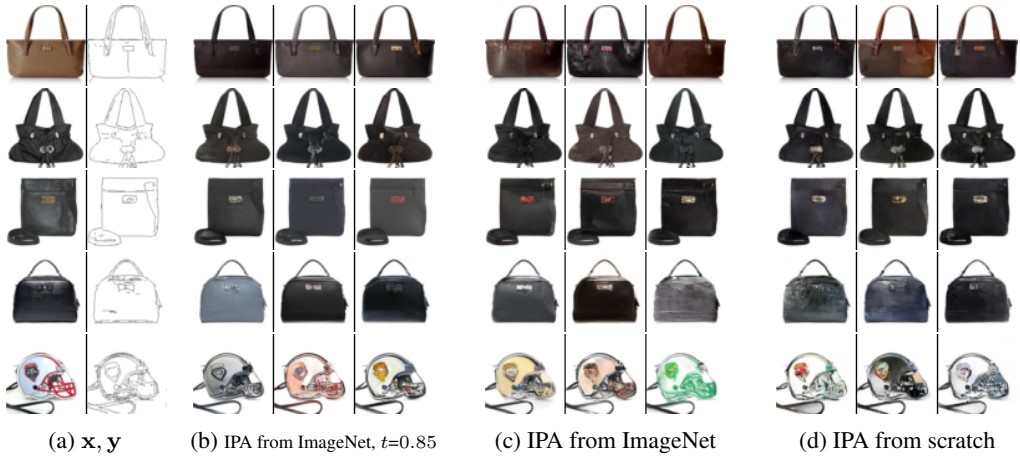

(a) $\mathbf{x}, \mathbf{y}$     (b) IPA from ImageNet, $t$=0.85     (c) IPA from ImageNet     (d) IPA from scratch

Figure 18: Comparison of conditional generations on Edges2Bags.

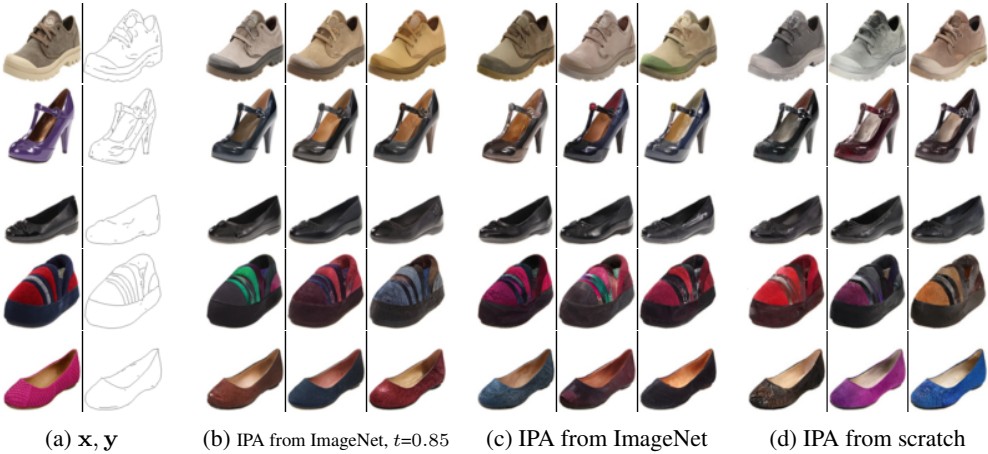

(a) $\mathbf{x}, \mathbf{y}$     (b) IPA from ImageNet, $t$=0.85     (c) IPA from ImageNet     (d) IPA from scratch

Figure 19: Comparison of conditional generations on Edges2Shoes.

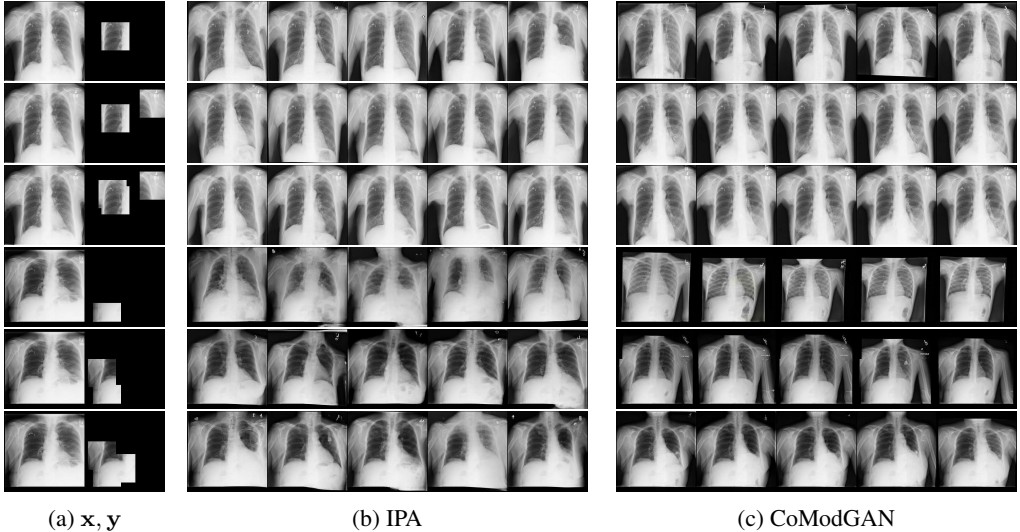

(a) $\mathbf{x}, \mathbf{y}$          (b) IPA          (c) CoModGAN

Figure 20: Sampled completions given three different masks for each of two ground truth x-ray images. For the first ground truth image (top three rows), there is little obvious difference between the completions produced by IPA and by CoModGAN. However, for each of the masks applied to the second ground truth image (bottom three rows), CoModGAN produces a posterior with minimal diversity, and which does not appear to include the ground truth. We inspected completion panels for CoModGAN on 20 different ground truth images, and found that this type of mode collapse occurred in 5 of them. IPA, in contrast, produces reliably diverse image completions for any $\mathbf{y}$ or ground truth image inspected. We do not show samples with a reduced temperature, as we did not use them for BOED on the basis that this could significantly reduce coverage of the posterior.

