# OpenReview forum: "Conditional Image Generation by Conditioning Variational Auto-Encoders"
_ICLR.cc/2022/Conference — ICLR 2022 Poster_

### Official Review · Reviewer_EhZn · 2021-11-01

**Correctness:** 3
**Technical Novelty And Significance:** 3
**Empirical Novelty And Significance:** 3
**Recommendation:** 8
**Confidence:** 4

**Main Review:**

- Motivation is clear. I especially like the third note in the introduction on the importance of this work: I believe that research that benefits from publicly-available pretrained weights (which are initially extremely costly to train) is a nice direction.
- A minor comment on the discussion in Section 3: Defining an expressive conditional model in which the decoder is capable of generating a plausible output without requiring the explicit contribution of the condition has an additional benefit of avoiding posterior collapse when the conditioning signal is strong enough such that the decoder can generate "a" good sample that minimizes the reconstruction loss.
- Regarding the Theorem 3.2, which only applies if the artifact are learned on the same dataset as the conditional VAE is trained on: Although it has been empirically shown that pretraining on a different dataset is still "useful", experiment in Fig. 4 suggests that the gap to the scenario where IPA pretrained on same dataset and on ImageNet is large, especially for the datasets that the domain gap to ImageNet is large. It would be great to have authors' view on feasible ways to bridge such gap when using pretrained artifacts?

Generally,
- Results are impressive, given the training budget; the method achieves near state-of-the-art performance in terms of visual fidelity and even better in terms of diversity.
- The paper is well-written and easy to follow

**Summary Of The Paper:**

This paper presents a CVAE-based approach for conditional image generation, in which a pretrained VAE is utilized to speed-up the training process. The paper clearly demonstrates the benefit of using a pretrained artifact in terms of time/resource of training as well as the image generation quality and diversity.

**Summary Of The Review:**

The paper presents a simple but well-articulated approach for conditional image generation, where one wants to avoid training the entire network from scratch. I believe the idea of using existing publicly available models is good, and this method neatly utilizes such available models to achieve impressive performance (and to avoid complication and challenges of training the full model).

---

> ### Author Response · Authors · 2021-11-17
> **Response to Reviewer EhZn**
>
> Thank you for your thoughtful comments and feedback. To address a couple of your points:
>
> ### Comment on posterior collapse
>
> This is a good point - by not feeding $\mathbf{y}$ to the decoder, we are “forcing” it to use the latent variables for conditioning which is likely to help avoid posterior collapse. We have __added a sentence to Section 6 mentioning this__.
>
> ### Feasible ways to improve cross-dataset performance of pretrained artifacts
>
> We can envisage two broad approaches: either fine-tuning some/all of the weights in $\theta$ and $\phi$, or using a more expressive partial encoder (i.e. using non-Gaussian $\hat{q}(z_l|z_{<l},\mathbf{y})$).
>
> The main advantage of finetuning $\theta$ and $\phi$ would be that it has a more comprehensive effect on the loss. Specifically, if we decompose the objective as
> $$ \mathcal{O}_\text{for}(\theta,\phi,\hat{\phi}) = \mathbb{E}_\{p_\text{data}(\mathbf{x},\mathbf{y})\} \left[ \underbrace{ \mathbb{E}_\{q(z|\mathbf{x};\phi)\} [ \log p (\mathbf{x}|z;\theta) ] }_\text{likelihood} - \underbrace{\text{KL}(q(z|\mathbf{x};\phi)||\hat{q}(z|\mathbf{y};\hat{\phi}))}_\text{KL divergence} \right], $$
> then fine-tuning the encoder/decoder weights can optimize both the likelihood and KL divergence terms. In contrast, improving the partial encoder can only improve the KL divergence term. The main disadvantage of tuning $\theta$ and $\phi$ is the increased time per iteration and memory footprint due to the additional gradients that must be calculated. Perhaps one way to obtain the advantage without the disadvantage would be to design VAE architectures for fast (and potentially gradient-free) adaptation, perhaps incorporating meta-learning techniques like the FiLM layers of CNAPS [1].
>
> Regarding our suggestion of using a more expressive partial encoder, we investigated using normalizing flows to parameterise $\hat{q}(z|\mathbf{y})$ and found that this slightly improved the ELBO but with little effect on any other metrics. We only experimented with this when same-dataset pretraining was used, so it is possible that it could have a bigger effect with cross-dataset pretraining.
>
> [1] Requeima, James, et al. "Fast and flexible multi-task classification using conditional neural adaptive processes." Advances in Neural Information Processing Systems 32 (2019): 7959-7970.

---

### Official Review · Reviewer_Fk3K · 2021-11-01

**Correctness:** 4
**Technical Novelty And Significance:** 4
**Empirical Novelty And Significance:** 4
**Recommendation:** 8
**Confidence:** 3

**Main Review:**

The paper presents a simple, but nice idea.  Being able to leverage foundational models clearly saves a huge amount of computer time.  The evaluation carried out is very extensive and the results persuasive.  The use case scenario seems reasonable.

I struggled with understanding how the conditional encoder is trained in such a way that the generated image is both diverse and accurate.  It clearly works and I may have just missed the explanation, but it is rather unclear to me what the objective described in Equation (7) does.  I am therefore going to score this lower than I would otherwise.  If you can clarify this either by showing me the explanation that I have over-read or by adding a short explanation (which given the page limit could be in the supplementary material), then I would be happy to reconsider and increase my score.

As a side remark, even the improved training times appear rather excessive.  Maybe this just a consequence of the difficulties of the problem.  It is though a bit sad.

**Summary Of The Paper:**

The paper proposes a method of leveraging state-of-the-art VAE decoders (foundational models) to create new conditional VAEs by training a new encoder for prior of the latent space.  The authors demonstrate quite convincingly that the new conditional VAE can generate images with more variation than a state-of-the-art GAN while achieving a comparable quality of image.  The advantage of the approach is that it substantially reduces training times (a drawback of traditional VAEs).  They then propose a task in the domain of X-ray imaging where the improved variation leads to improved performance in targetted X-ray measurements.

**Summary Of The Review:**

Overall this is a strong paper.  I am supportive of seeing this published.  I am missing an explanation of the authors' objective function and why it should find a diverse, but accurate prior.  This may just be my stupidity, but I think a clear explanation of this might be helpful to other readers (apologies, if I just read over this, but it wasn't obvious to me reading the main paper).  I am happy to increase my score if I can be satisfied on this point.

In view of your answer I have increased your score.

---

> ### Author Response · Authors · 2021-11-17
> **Response to Reviewer Fk3K**
>
> Thank you for your helpful feedback. We’ll try to address your concern regarding why the generated images are diverse and accurate. For simplicity, in this discussion we will assume that we want to predict all dimensions of $\mathbf{x}$ (meaning that $\mathbf{\tilde{x}}:=\mathbf{x}$ in the notation introduced in response to Reviewer 67T5).
>
> Our argument centres on the “mass-covering” properties of the forward KL divergence. When a variational distribution (e.g. $p_\text{cond}(\mathbf{x}|\mathbf{y})$) is fitted to a reference distribution (e.g. $p_\text{data}(\mathbf{x}|\mathbf{y)}$) using the forward KL divergence ($\text{KL}( p_\text{data}(\mathbf{x}|\mathbf{y)} || p_\text{cond}(\mathbf{x}|\mathbf{y}) )$), it is heavily penalised for assigning low probability density to any regions of high density under the reference distribution. The variational distribution is therefore forced to assign probability mass broadly over the reference distribution and, if it cannot fit the reference distribution perfectly, to typically have slightly higher variance than the reference distribution. For further explanation, see e.g. this blog post: https://www.tuananhle.co.uk/notes/reverse-forward-kl.html.
>
> Therefore, in order to obtain a mass-covering (and hence diverse) $p_\text{cond}(\mathbf{x}|\mathbf{y})$, we want to fit a mass-covering KL divergence to $p_\text{data}(\mathbf{x}|\mathbf{y)}$. Such a KL divergence is difficult to estimate, but Theorem 3.1 suggests that optimising $\mathcal{O}_\mathrm{for}$ is equivalent to optimising a similar mass-covering KL divergence in latent space:
> $$    \text{argmax}_\{\hat{\phi} \in \hat{\Phi{}}\} \mathcal{O}_\mathrm{for}(\theta, \phi, \hat{\phi}) = \text{argmin}_\{\hat{\phi} \in \hat{\Phi{}}\} \mathbb{E}_\{p_\text{data}(\mathbf{y})\} \left[ \text{KL}( r(z|\mathbf{y}; \phi) || \hat{q}(z|\mathbf{y}; \hat{\phi}) ) \right]. $$
> Here $r(z|\mathbf{y}; \phi)$ plays the role of the reference/data distribution and $\hat{q}(z|\mathbf{y}; \hat{\phi})$ is our learned distribution in latent space. Due to the mass-covering KL divergence, $\hat{q}(z|\mathbf{y}; \hat{\phi})$ should learn a diverse distribution over $z$. Intuitively, this should lead to a diverse distribution over the images $\mathbf{x} \sim p_\text{model}(\cdot|z)$ when $z\sim \hat{q}(\cdot|\mathbf{y}; \hat{\phi})$.
>
> The above relies on a slight leap of reasoning between having diverse samples of $z$ and having diverse samples of $\mathbf{x}$. More formally, as __we now show in Appendix C.2__, we can argue that we are minimising an upper-bound on the KL-divergence in $\mathbf{x}$-space:
> $$ \mathbb{E}_\{p_\text{data}(\mathbf{y})\} \left[ \text{KL}( p_\text{data}(\mathbf{x}|\mathbf{y}) || p_\text{cond}(\mathbf{x}|\mathbf{y}) ) \right] \leq \mathbb{E}_\{p_\text{data}(\mathbf{x},\mathbf{y})\} \left[ \log p_\text{data}(\mathbf{x}|\mathbf{y)} \right] - \mathcal{O}_\mathrm{for}(\theta,\phi,\hat{\phi}).  $$
> Maximising $\mathcal{O}_\mathrm{for}(\theta,\phi,\hat{\phi})$ will therefore minimise an upper bound on the mass-covering KL divergence of interest, $\text{KL}( p_\text{data}(\mathbf{x}|\mathbf{y}) || p_\text{cond}(\mathbf{x}|\mathbf{y}) )$, on expectation over $p_\text{data}(\mathbf{y})$.
>
> In the submitted version of the paper, we made a considerably shorter version of this argument in the __paragraph after Theorem 3.1. We have reworded this to hopefully make it clearer, and now reference the bound on the KL divergence in $\mathbf{x}$-space, which we derive in Appendix C.2.__ We hope this helps to address your concern. Please let us know if there is anything else we can do to clarify this point.

---

### Official Review · Reviewer_gc2C · 2021-11-01

**Correctness:** 4
**Technical Novelty And Significance:** 2
**Empirical Novelty And Significance:** 2
**Recommendation:** 6
**Confidence:** 3

**Main Review:**

Strengths:

1. The paper has strong conceptual and mathematical motivation and has firm theoretical foundations.

2. The approach makes intuitive sense; it follows that pieces of an unconditional generative model can be used to build a conditional one.

3. The method demonstrates promising applications. Image completion is crucial for computational photography, and application to medical imaging is an additional route to wide applications in society.

Weaknesses:

1. Technical and empirical novelty is limited; this is effectively a small modification to unconditional VAEs to create a conditional version. In addition, while quantitative results are competitive with respect to GANs, the performance margin is not particularly exceptional.

2. The structure and flow of the paper could be improved. For example, the related works section is on one of the last pages of the paper.

3. The choice of datasets is fairly simplistic (CIFAR-10, FFHQ-256, Edges2Shoes, Edges2Handbags). While there is pretraining on the ImageNet dataset, it would strengthen the paper to evaluate on the ImageNet dataset in a manner like CIFAR-10 and FFHQ-256.

**Summary Of The Paper:**

In this paper, the authors focus on training conditional variational autoencoders. They propose an architecture and training objective which leverages pretraining an initial unconditional VAE. The approach effectively infers the latent variables of the original unconditional VAE given the new conditioning input. They demonstrate the method on a number of datasets, specifically focusing on image completion. They show competitive performance with adversarial approaches. They also demonstrate an application in the realm of medical imaging.

**Summary Of The Review:**

The authors propose a modification to variational autoencoders which improve them in a conditional setting. The show promising results on image completion on a number of datasets and also in the area of medical imaging. However, the paper suffers from a few drawbacks. First, the technical and empirical novelty is limited. Second, the choice of datasets used for evaluation is simplistic.

In light of the other reviewers concerns and the response from the authors, I upgrade my score.

---

> ### Author Response · Authors · 2021-11-17
> **Response to Reviewer gc2C**
>
> Thank you for your helpful comments. We have updated the manuscript with ImageNet experiments as suggested, and will here address each of your points in turn.
>
> ### 1 - Novelty
>
> We argue that, while our performance is merely “competitive” with regard to FID scores, we do show significantly better coverage of the “true” posterior $p_\text{data}(\mathbf{x}|\mathbf{y})$. We say this based on both the reported LPIPS-GT scores and our qualitative results (see e.g. the failure mode of CoModGAN explored in Figure 13 in the appendix).
>
> ### 2 - Structure of paper
>
> We chose to put the related work section near the end of the paper in order to avoid breaking the flow of the paper (by attempting to relate our method to related approaches before having described it or by, e.g., splitting our discussion of VAEs in Section 2 from our discussion of the conditional VAE objective in Section 3). Positioning the related work in this way is not unusual: looking through the NeurIPS paper awards for the past 2 years, [1,2] have related work sections immediately before the conclusion. We reference various other high-impact papers with this structure [3-7].
>
> Having said that, we are of course keen to make the paper as readable as possible and take your concerns into account. We’re not sure exactly what issues resulted from our placement of the related work section, but if it would be helpful to have more context before seeing the experiments then one reasonable option might be to position the related work between the current Section 3 and Section 4. Would this help to address your criticism? Otherwise please let us know if there is anything else we can do.
>
> ### 3 - ImageNet experiments
>
> Thank you for this suggestion. We have begun experiments on ImageNet at 64x64 resolution, and results can be seen in __Figure 4__. The preliminary results are positive for IPA: the ELBO for IPA is clearly better than the ELBO for a CVAE trained from scratch, and by a larger margin than for the other datasets plotted. We are currently training CoModGAN on this dataset as a baseline, and will add this comparison when it has finished (expected in three days).
>
> [1] (NeurIPS 2019) Löwe, Sindy, Peter O'Connor, and Bastiaan S. Veeling. "Putting an end to end-to-end: Gradient-isolated learning of representations."
>
> [2] (NeurIPS 2020) Brown, Tom B., et al. "Language models are few-shot learners."
>
> [3] (NeurIPS 2019) Rajeswaran, Aravind, et al. "Meta-learning with implicit gradients."
>
> [4] (NeurIPS 2020) Ho, Jonathan, Ajay Jain, and Pieter Abbeel. "Denoising diffusion probabilistic models."
>
> [5] (NeurIPS 2019) Song, Yang, and Stefano Ermon. "Generative modeling by estimating gradients of the data distribution."
>
> [6] (ICLR 2018) Song, Yang, et al. "Pixeldefend: Leveraging generative models to understand and defend against adversarial examples."
>
> [7] (NeurIPS 2019) Ilyas, Andrew, et al. "Adversarial examples are not bugs, they are features."
>
> EDIT after further revision: Figure 12 -> Figure 13

---

> > ### Author Response · Authors · 2021-11-23
> > **Updated ImageNet-64 results in manuscript**
> >
> > Since this is the last day for paper revisions to be made, we have made a final update to the ImageNet-64 results presented. To summarise our most significant results:
> > - In Table 1 we compare final FID, P-IDS, and LPIPS-GT scores for IPA (after 5 days of training on 1 GPU) with those achieved by CoModGAN (trained for 138 hours on 4 GPUs to reach the recommended 25 million iterations). IPA outperforms CoModGAN on all metrics.
> > - We compare IPA on ImageNet-64 with a from-scratch baseline in Figure 4. Even after more than a week of training, the from-scratch baseline does not reach the FID or ELBO achieved by IPA within 1 day. IPA’s improvement compared to the from-scratch baseline is as large as, or larger than, that for the previous datasets considered.
> >
> > We present some further results in the appendix:
> > - Figure 6 plots multiple evaluation metrics against the mask distribution on datasets including ImageNet-64.
> > - Figure 7 shows the faithfulness weighted variance metric (suggested by Reviewer 67T5) on datasets including ImageNet-64.
> > - Figure 25 shows samples from IPA and the from-scratch baseline on ImageNet-64.
> >
> > For completeness we will run additional baselines on ImageNet-64 (PIC and IPA-R). We will not be able to obtain these results before the end of the revision period, but do not expect them to be competitive with the results shown for IPA and CoModGAN.
> >
> > Thanks again for your suggestions. We hope that the positive results we’ve presented on ImageNet-64, along with our replies to your other points, have helped to address your concerns. Please let us know if there is anything else we can do or discuss, and consider raising your final score if not. Thank you.

---

### Official Review · Reviewer_67T5 · 2021-11-02

**Correctness:** 3
**Technical Novelty And Significance:** 3
**Empirical Novelty And Significance:** 3
**Recommendation:** 8
**Confidence:** 5

**Main Review:**

Strong Points:

1. Reusing pretrained models that cost huge amount of computing is a well-motivated problem, especially for models like VAEs that take a very long time to train
2. The experiments contain enough details and the comparisons to the baselines are comprehensive. The results also support the claim that the proposed method could achieve descent performance given a tight computation budget
3. The comparisons to CoModGAN on completion tasks with very little observed input area are very interesting

Weak Points:

1. The model is based on the hierarchical VAE framework, therefore, it should be made clear that it is not directly minimizing the KL divergence of $r(z|y;\phi)$ and $\hat{q}(z|y;\hat{\phi}))$, rather, it should be the KL divergence of $r(z_i|y;\phi)$ and $\hat{q}(z_i|y;\hat{\phi}))$ since $\hat{q}(z|y;\hat{\phi}))$ is a multiplication of a chain of Gaussian distributions which itself is intractable as well. An example in the unconditional case could be found in equation 1 in [a]
2. The adopted diversity metric, LPIPS-GT, is not ideal here. LPIPS-GT would be a good metric for measuring the mode coverage of the training targets, but cannot capture diversity that well. Consider a case of generating 100 samples, having 1 sample far away (e.g. LPIPS distance 0.1) and 99 close to the target (e.g. LPIPS distance 0.01) vs. generating 100 samples whose distances are uniformly spread across a given range (e.g. LPIPS distance 0.01 ~ 0.1). Clearly, the latter is more diverse but the two scenarios will have the same LPIPS-GT score.  Alternative diversity measurement: LPIPS diversity score [b], faithfulness-weighted variance [c]
3. The formulation of partial VAE whose decoder does not depend on the conditional input $y$ is not new [d]. Though the authors pointed out    that the focus in that paper is different, the novelty in the technical formulation is not significant. Therefore, I gave a 2 on "Technical Novelty And Significance"
4. In appendix B2: how do you get from equation (16) to (17)?
5. In appendix B4, equation (33), the second Expectation should be $\hat{q}(z|y)$ instead of $\hat{q}(z|x)$?


References

[a] Arash Vahdat and Jan Kautz. Nvae: A deep hierarchical variational autoencoder. *arXiv preprint
arXiv:2007.03898*, 2020.

[b] Zhu, Jun-Yan, Richard Zhang, Deepak Pathak, Trevor Darrell, Alexei A. Efros, Oliver Wang and Eli Shechtman. “Toward Multimodal Image-to-Image Translation.” *NIPS* (2017).

[c] Li, Ke, Shichong Peng, Tianhao Zhang and Jitendra Malik. “Multimodal Image Synthesis with Conditional Implicit Maximum Likelihood Estimation.” *International Journal of Computer Vision*(2020): 1-22.

[d] Chao Ma, Sebastian Tschiatschek, Konstantina Palla, Jose ́ Miguel Herna ́ndez-Lobato, Sebastian Nowozin, and Cheng Zhang. Eddi: Efficient dynamic discovery of high-value information with partial vae. arXiv preprint arXiv:1809.11142, 2018.

**Summary Of The Paper:**

This paper proposes a method, IPA, that converts an unconditional VAE to a conditional one by reusing the pretrained weights and training a partial encoder. Experiments on the image completion task show favorable results compared to GAN-based approach. The authors also explored an application in Bayesian optimal experimental design which takes advantage of the the posterior estimation of IPA.

**Summary Of The Review:**

This paper is well-structured and contains enough detail overall. The problem setting is well-motivated and the proposed method showed promising results towards addressing the issue. However, the concerns listed above should be addressed to make the submission sound and complete.

---

> ### Author Response · Authors · 2021-11-17
> **Response to Reviewer 67T5 (1/3)**
>
> Thank you for your very thorough review. We will address each of your concerns in turn and describe how we have updated the paper accordingly.
>
> ### 1 - “not directly minimizing the KL divergence of $r(z|y;\phi)$ and $\hat{q}(z|y;\hat{\phi})$"
> This is a good point and we have __edited Section 3.1 to make it clearer that we estimate the KL divergence as in [a]__, by summing the KL divergence for each layer, computed conditioned on Monte Carlo samples of the higher-level latent variables. However, we also wish to emphasise that this is an unbiased estimate of the intractable KL divergence between the chains of Gaussian distributions. To show this in the unconditional case:
>
> $$
> KL(q(z|\mathbf{x}) || p(z)) = \mathbb{E}_{q(z|\mathbf{x})}\left[ \log \frac{q(z|\mathbf{x})}{p(z)} \right]
> $$
>
> $$
> KL(q(z|\mathbf{x}) || p(z)) = \mathbb{E}_{q(z|\mathbf{x})}\left[ \log \frac{\prod_\{l=1\}^L q(z_l|z_\{\<l\},\mathbf{x})}{\prod_\{l=1\}^Lp(z_l|z_\{\<l\})} \right]
> $$
>
> $$
> KL(q(z|\mathbf{x}) || p(z)) = \mathbb{E}_{q(z|\mathbf{x})}\left[ \sum_\{l=1\}^L \log \frac{q(z_l|z_\{\<l\},\mathbf{x})}{p(z_l|z_\{\<l\})} \right]
> $$
>
> $$
> KL(q(z|\mathbf{x}) || p(z)) =  \sum_\{l=1\}^L \mathbb{E}_{q(z_\{\leq l\}|\mathbf{x})}\left[ \log \frac{q(z_l|z_\{\<l\},\mathbf{x})}{p(z_l|z_\{\<l\})} \right]
> $$
>
> $$
> KL(q(z|\mathbf{x}) || p(z)) =  \sum_\{l=1\}^L \mathbb{E}_{q(z_\{\<l\}|\mathbf{x})}\left[ \text{KL}(q(z_l|z_\{\<l\},\mathbf{x}) || p(z_l|z_\{\<l\}) ) \right]
> $$
>
> where $z_\{\<1\}$ is defined as the null set. Or, as written in equation 1 of [a],
>
> $$
> KL(q(z|\mathbf{x}) || p(z)) = \text{KL}(q(z_1|\mathbf{x}) || p(z_1) ) + \sum_\{l=2\}^L \mathbb{E}_{q(z_\{\<l\}|\mathbf{x})}\left[ \text{KL}(q(z_l|z_\{\<l\},\mathbf{x}) || p(z_l|z_\{\<l\}) ) \right].
> $$
>
> We can use the same argument for the KL divergence which we compute as part of our objective $\mathcal{O}_\mathrm{for}$:
>
> $$
> KL(q(z|\mathbf{x}) || \hat{q}(z|\mathbf{y})) = \text{KL}(q(z_1|\mathbf{x}) || \hat{q}(z_1|\mathbf{y}) ) + \sum_\{l=2\}^L \mathbb{E}_{q(z_\{\<l\}|\mathbf{x})}\left[ \text{KL}(q(z_l|z_\{\<l\},\mathbf{x}) || \hat{q}(z_l|z_\{\<l\},\mathbf{y}) ) \right].
> $$
>
> In practice, we estimate the expectations on the right-hand side with Monte Carlo samples, yielding an unbiased estimate of the KL divergence on the right-hand side. We therefore believe that it is reasonable to describe this procedure as "directly minimizing" this KL divergence, despite the fact that we never compute it exactly. __We have added the above derivation to the appendix.__

---

> ### Author Response · Authors · 2021-11-17
> **Response to Reviewer 67T5 (2/3)**
>
> ### 2 - Diversity metrics
>
> We have computed the diversity scores that you referenced and included them in the appendix, but we argue that the LPIPS-GT is a better metric for what we are trying to investigate. Rather than simply producing diverse samples, we want to produce samples from a distribution with "calibrated diversity". If $\mathbf{y}$ is uninformative, then a well-calibrated distribution will be uncertain, and samples from it will be diverse. On the other hand, if $\mathbf{y}$ is very informative, there may be little to no variation in samples from a well-calibrated $p_\text{cond}(\mathbf{x}|\mathbf{y})$. For applications like the Bayesian optimal experimental design example in Section 5, this type of well-calibrated uncertainty is required rather than simply diversity. To clarify this point, we have slightly __reworded the third paragraph of Section 4__.
>
> Given that you say the LPIPS-GT metric measures the “mode coverage of the training targets”, we wish to clarify that we compute it on the test set and not the training set. The only way for a trained model to ensure that all modes of a held-out test set are covered is to cover all modes of the true posterior $p_\text{data}(\mathbf{x}|\mathbf{y})$ for any $\mathbf{y}$ (i.e. be at least as diverse as the true posterior). At the same time, the LPIPS-GT will penalise models for being excessively diverse: if samples are spread across a larger space then fewer of them will be contained in the same mode as the ground-truth and so, on expectation, the minimum LPIPS distance to the ground truth will be larger. By penalising methods for being either too diverse or insufficiently diverse, we believe that the LPIPS-GT provides a reasonable metric for how well-calibrated $p_\text{cond}(\mathbf{x}|\mathbf{y})$ is.
>
> The LPIPS diversity score [b] is a good measure of diversity, but not a good measure of how well calibrated this diversity is. No matter how informative $\mathbf{y}$, and how concentrated the posterior $p_\text{data}(\mathbf{x}|\mathbf{y})$, the LPIPS diversity score is maximised by making the sampled images very diverse. This issue is related to that noted by [b] when they introduced the LPIPS diversity score: good "scores may also indicate that unnatural images are being generated, causing meaningless variations".
>
> The faithfulness weighted variance (FWV) measures a combination of diversity and similarity to the ground-truth. However, it requires computing LPIPS distances between image samples and pixel-space averages of samples. Since pixel-space averages are not generally valid images themselves, it is not clear whether these LPIPS distances are meaningful in our case. This is different to the case in [c], where the FWV is used to evaluate super-resolution (for which image samples may be more similar to eachother and hence pixel-space averages more reasonable). Therefore, although IPA performs well according to the FWV metric, we find it less intuitive than LPIPS-GT and are not sure how much to trust it. We have therefore only plotted it in the appendix. We are open to moving it if you disagree with this assessment.
>
> Nonetheless, we present the LPIPS diversity score, LPIPS-GT, and faithfulness-weighted variances in the following tables for image completion on 3 datasets. We use them to compare IPA, CoModGAN (with the caveat that it is not yet fully trained on ImageNet), and a baseline unconditional VD-VAE (Child 2020, cited in paper). When sampling, observed pixels are fixed to their observed values (as in all scores in the paper). Image completions produced by the unconditional VD-VAE are therefore almost never coherent. Despite this, because they completely ignore $\mathbf{y}$, samples from the unconditional VAE are very diverse. As a result, they obtain better LPIPS diversity scores than either IPA or CoModGAN. We believe this shows a major flaw in using the LPIPS diversity score to evaluate how well calibrated a method's diversity is. For FWV, IPA is the best-performing method. There is no clear winner between CoModGAN and the unconditional VAE, and we maintain that it is not clear how meaningful the FWV metric is in our case. __We have nonetheless added the computed FWV and LPIPS diversity scores to the appendix.__
>
> ### LPIPS-GT (lower is better)
> |   | CIFAR-10 | ImageNet-64 | FFHQ-256 |
> |---|---|---|---|
> | IPA | **0.0262** | **0.1396** | **0.123** |
> | CoModGAN | 0.0326 | 0.1662 | 0.143 |
> | Unconditional VAE | 0.0516 | 0.2225 | 0.283 |
>
> ###  LPIPS diversity score [b] (higher is better)
> |   | CIFAR-10 | ImageNet-64 | FFHQ-256 |
> |---|---|---|---|
> | IPA | 0.0473 | 0.1924 | 0.1334 |
> | CoModGAN | 0.0662 | 0.2435 | 0.0979 |
> | Unconditional VAE | **0.0892** | **0.2949** | **0.2901** |
>
>
> ### Faithfulness weighted variance [c] (higher is better)
> |   | CIFAR-10 | ImageNet-64 | FFHQ-256 |
> |---|---|---|---|
> | IPA | **3.004** | **1.826** | **2.426** |
> | CoModGAN | 1.693 | 0.8563 | 1.239 |
> | Unconditional VAE | 2.361 | 0.6863 | 0.5907 |

---

> ### Author Response · Authors · 2021-11-17
> **Response to Reviewer 67T5 (3/3)**
>
> ### 3 - Technical novelty
>
> We would like to point out another difference from [d] (which we should have stated explicitly in the related work section, __and now do__): we use a different training objective. We describe [d]’s training objective in Appendix E, and show experiments with it using the acronym IPA-R (which is generally outperformed by IPA). Therefore, we would argue that the combination of our conditional VAE formulation and training objective is novel (as well as our use of pretrained weights).
>
> ### 4 - Proof of Theorem 3.2
>
> Thank you for catching this! We have realised there was an additional assumption required to make this step: that $p^*(\mathbf{x}|z)=p^*(\mathbf{x}|z,\mathbf{y})$. In other words, that under $p^*$, $\mathbf{x}$ and $\mathbf{y}$ are conditionally independent given $z$. If this is the case, then we could write the numerator of Eq. (16) (in the old version of the paper) as
> $$ p^*(\mathbf{x}|z)p^*(z|\mathbf{y})=p^*(\mathbf{x}|z,\mathbf{y})p^*(z|\mathbf{y})=p^*(\mathbf{x},z|\mathbf{y}) = p^*(z|\mathbf{x},\mathbf{y})p^*(\mathbf{x}|\mathbf{y})=p^*(z|\mathbf{x})p^*(\mathbf{x}|\mathbf{y}) $$
> which is the numerator of Eq. (17) (in the old version of the paper).
> The last step here follows because we defined $p^*(z|\mathbf{x},\mathbf{y})=q(z|\mathbf{x};\phi^*)=p^*(z|\mathbf{x})$. The proof for Theorem 3.2 would therefore hold.
>
> In the image completion case, the assumption that $p^*(\mathbf{x}|z)=p^*(\mathbf{x}|z,\mathbf{y})$ will hold if $\mathbf{x}$ and $\mathbf{y}$ consist of disjoint sets of image pixels, e.g., $\mathbf{x}$ denotes the unobserved pixel values and $\mathbf{y}$ denotes the observed values. This is because the VAE we use has a likelihood which is pixel-wise independent, making any disjoint sets of pixels conditionally independent given $z$. However, we had defined $\mathbf{x}$ to be the entire image, including pixels observed in $\mathbf{y}$, and so the assumption would not hold. To rectify this, we have rewritten the training objective to only include the likelihood of $\mathbf{\tilde{x}}$: for the case of image completion, we define $\mathbf{\tilde{x}}$ to consist purely of observed pixels. Informally, $\mathbf{\tilde{x}} := \mathbf{x} \setminus \mathbf{y}$. The assumption required for Theorem 3.2 then becomes $p^*(\mathbf{\tilde{x}}|z)=p^*(\mathbf{\tilde{x}}|z,\mathbf{y})$. Since $\mathbf{\tilde{x}}$ and $\mathbf{y}$ are defined to consist of disjoint sets of image pixels, this will be satisfied. Importantly, training IPA using the modified objective with $\mathbf{\tilde{x}}$ is identical to training with IPA with the original training objective. This is due to the fact that IPA learns only $\hat{\phi}$. The modified likelihood term only has non-zero gradients with respect to $\theta$. We therefore do not, in practice, compute this likelihood term while training IPA, and IPA's training dynamics are identical between the original and reformulated objective.
>
> __We have added this assumption to the statement of Theorem 3.2__ by writing
> $$I^*_\{\mathbf{\tilde{x}},\mathbf{y}|z\} := \mathbb{E}_\{p_\text{model}(\mathbf{x}, \mathbf{y}, z; \theta^*)\} \left[ \log \frac{p_\text{model}(\mathbf{\tilde{x}}|z, \mathbf{y}; \theta^*) }{ p_\text{model}(\mathbf{\tilde{x}}|z; \theta^*) } \right] = \mathbb{E}_\{p^*(\mathbf{\tilde{x}}, \mathbf{y}, z)\} \left[ \log \frac{p^*(\mathbf{\tilde{x}}|z, \mathbf{y}) }{ p^*(\mathbf{\tilde{x}}|z) } \right] = 0.$$
>
> For tasks (other than inpainting) where $I^*_{\mathbf{\tilde{x}},\mathbf{y}|z}$ may not be zero, $I^*_{\mathbf{\tilde{x}},\mathbf{y}|z}$ provides a bound on the difference between the left- and right-hand sides of the equation in Theorem 3.2. __This is shown in Equation (41) and discussed in Appendix D.2.__ We also argue in Appendix D.2 that $I^*_{\mathbf{\tilde{x}},\mathbf{y}|z}$ will be "close" to zero when $\mathbf{y}$ represents high-level features, as is the case in our Edges2Photos experiments.
>
> Also, note that while addressing this point we have restructured the section of the appendix containing the proofs, so section numbers and equation numbers have changed. __The proof of Theorem 3.2 is now in Appendix D.1, and the step that you originally took issue with is analogous to the step between Equations (27) and (32)__, with additional intermediate steps shown in between.
>
> ### 5 - Typo
>
>  Good catch, thank you!

---

> > ### Comment · Reviewer_67T5 · 2021-11-21
> > **Re: Response to Reviewer 67T5**
> >
> > Thank you for the response!
> >
> > A comment on response of 2: I think the adjusted wording for LPIPS GT which measures calibration to $p_{cond}(x|y)$ is a better fit for the problem setting and introduces less ambiguities. Estimating diversity in an ill-posed problem setting (especially for datasets that contain only one observed image for each input) usually involves finding a balance on a spectrum: on one end, it penalizes samples for not being close to the observed image and on the other end, the criteria is more relaxed and every sample is counted more evenly (and this is what the bandwidth parameter $\sigma$ is controlling in the faithfulness-weighted variance).
> >
> > Regarding section D.2 in the appendix: I think the claim that "$I^*_{\tilde{x},y|z}$ is very small" is very task dependent and may not hold true for some conditional image generation tasks, for example, in super-resolution this might not hold well since if you take $\tilde{x}$ to be the difference of a blurred image y and the observed image x, then the mutual information of the two might be high. Since the method is presented as a conditional image generation method, it would be more informative to the readers to discuss the conditions where that assumption may not hold or list it as a potential limitation (at least in terms of theoretical guarantees).

---

> > > ### Author Response · Authors · 2021-11-21
> > > **Response**
> > >
> > > Thanks for the helpful comments on our revisions.
> > >
> > > Regarding your comment on point 2, we are glad that the wording now more accurately conveys how we wish to evaluate the method. Given your comments, we also agree that comparing the FWV for various values of $\sigma$ can provide more insight than reporting a single number. We have therefore added Figure 7 in the appendix, presenting the FWV for various values of $\sigma$ (those reported by [c], two higher values, and two lower values). We find that IPA achieves the highest FWV for almost every combination of dataset, $\sigma$, and mask distribution (although is sometimes outperformed by IPA-R on CIFAR10).
> > >
> > > We agree with your comments on Appendix D.2. We didn’t claim that $I^*_{\tilde{x},y|z}$ is small in general, and have added some extra discussion to make clear that $I^*_{\tilde{x},y|z}$ will be:
> > > - zero for inpainting
> > > - likely small for conditioning on high-level features such as in edges-to-photos or class-conditional generation,
> > > - and potentially large for e.g. super-resolution and image colourisation.
> > >
> > > In particular, we have added this discussion in the first 4 lines of the final paragraph of Section 3, the 6th-9th line of Section 7, and the final 4 lines of the first paragraph of Appendix D.2.
> > >
> > > Thanks again for your insightful feedback and discussion. We think that the changes made have strengthened the paper and hope that they addressed your concerns. If you have any further comments, we would be happy to discuss them and make appropriate changes to the paper (if suggested before the revision period ends). Otherwise, please consider raising your final rating. Thank you very much.

---

> > > > ### Comment · Reviewer_67T5 · 2021-11-22
> > > > **Raising Score To 8**
> > > >
> > > > Thank you for the response and addressing my concerns. Therefore, I raise my score to 8.

---

### Author Response · Authors · 2021-11-17
**Overall response**

We thank the reviewers for their thoughtful and helpful comments. We will reply to each separately, but here summarise the main revisions we have made to the paper in response:

- We have added experiments with ImageNet-64, as suggested by Reviewer gc2C. We have compared IPA with CoModGAN and a conditional VAE trained from scratch, with IPA generally outperforming the other methods across all metrics. Please see [our response to Reviewer gc2c](https://openreview.net/forum?id=7MV6uLzOChW&noteId=WZUPnaSAi0l) for more details. ~~We are running new experiments with ImageNet, as suggested by Reviewer gc2C. Although the baselines for these experiments are not yet complete, we have updated Figure 4 with the ELBO and FID scores throughout training so far. It is apparent that using IPA provides substantial advantages over training from scratch, perhaps even more so than for the other datasets plotted. We also show samples in the appendix (Figure 25). We are currently training CoModGAN (our strongest non-VAE baseline); we include preliminary results in the appendix and will revise the paper with a final version of this comparison before the revision period is over.~~
- Reviewer 67T5 pointed out an issue with the proof of Theorem 3.2, which required an additional assumption to fix (see our response to Reviewer 67T5 for more detail). Showing that this assumption is reasonable when IPA is used for image completion prompted us to introduce a new piece of notation, $\mathbf{\tilde{x}}$, denoting the part of $\mathbf{x}$ that is predicted by the VAE. For image completion, we define $\mathbf{\tilde{x}}$ to consist only of the “missing” pixels but, in other cases, we can use $\mathbf{\tilde{x}} := \mathbf{x}$. Given this definition, we define $\mathcal{O}_\mathrm{for}$ as a lower-bound on the likelihood of $\mathbf{\tilde{x}}$ given $\mathbf{y}$. In practice, however, this only affects gradients of the objective with respect to $\theta$. Since $\theta$ is not learned while training IPA, this change does not affect any of our experiments with IPA.
- We have added two additional measures of sample diversity to the appendix. See our response to Reviewer 67T5 for more details.
- Following the ICLR author guide, we now include the supplementary text in the same file as the paper.

EDIT after further revision: Figure 24 -> Figure 25

---

### Decision · Program_Chairs · 2022-01-20

**Decision:**

Accept (Poster)

**Comment:**

This paper presents a method to turn a pretrained unconditional VAE into a conditional VAE by training an encoder to predict the unconditional VAE latents given conditional input. On a variety of image tasks, the method is shown to perform competitively with GANs, yielding good sample quality and diversity, and resulting in training time that improves on direct conditional generation approaches. While the technical novelty is limited, the strong empirical results and relevance given the growing availability of pretrained unconditional models lead me to recommend accepting this paper.

Ethics concerns have been raised for this paper. In particular, there were concerns with respect to the application of generative models, which inherit biases from the dataset, to guide medical imaging. It would be good to discuss this issue in more depth. A second point that was raised by the ethics committee is the fact that chest X-rays are usually not taken in a sequential manner. We ask the authors to either provide evidence that X-rays can be taken sequentially (one can think of situations where that's the case, e.g., X-rays of teeth in the mouth), preferably in the context of chest X-rays; if that's not possible, please highlight that the application, as described in the paper, is unrealistic (at the moment), and that it only serves as an illustration.

The key point we therefore ask the authors to address is to ensure that the paper clearly states how realistic the application is and what potential problems may arise when using generative models in this particular domain.